# Evaluating the Diversity and Quality of LLM Generated Content

**Alexander Shypula**[1]    **Shuo Li**[1]    **Botong Zhang**[1]
**Vishakh Padmakumar**[2]    **Kayo Yin**[3]    **Osbert Bastani**[1]
[1]University of Pennsylvania, [2]New York University, [3]UC Berkeley
shypula@seas.upenn.edu

## Abstract

Recent work suggests that preference-tuning techniques—such as Reinforcement Learning from Human Preferences (RLHF) methods like PPO and GRPO, as well as alternatives like DPO—reduce diversity, creating a dilemma given that these models are widely deployed in applications requiring varied outputs. We argue that diversity without consideration of quality has limited practical value. To address this issue, we introduce a framework for measuring *effective semantic diversity*—diversity among outputs that meet quality thresholds—which better reflects the practical utility of large language models (LLMs). Using open-ended tasks that require no human intervention, we find counterintuitive results: when using diversity metrics that do not explicitly consider quality, preference-tuned models—particularly those trained via RL—often produce outputs with lower diversity; however, these same preference-tuned models generate greater effective semantic diversity than SFT or base models. Our analysis further shows another trend: while larger models may exhibit greater effective semantic diversity than smaller models, the smaller models are consistently more parameter-efficient at producing unique content within a fixed sampling budget. These findings have practical implications for applications that require diverse yet high-quality outputs, from creative assistance to synthetic data generation.

## 1  Introduction

As large language models (LLMs) increasingly serve as tools for ideation, synthetic data generation, and creative assistance, their ability to produce *diverse yet high-quality* outputs has become critically important. While recent advances have dramatically improved model performance, relatively little attention has been paid to systematically measuring and optimizing for the dual objective of quality and diversity.

Consider a user asking an LLM to generate story premises or a researcher using an LLM to create synthetic training data. In these scenarios, the utility of the model depends not only on producing a coherent set of responses but also on generating outputs that span a meaningful range of ideas or examples. This contrasts with traditional LLM evaluation paradigms, which often optimize for a single correct answer.

The challenge lies in defining what constitutes meaningful diversity. Diversity without quality is trivial to achieve—random tokens are maximally diverse but utterly useless. What is needed instead is diversity among outputs that meet a threshold of quality or acceptability. We refer to this as *effective semantic diversity* and argue that it is a more accurate measure of an LLM's practical utility in open-ended generation tasks.

This distinction is particularly relevant in today's landscape of post-training LLMs, where preference-tuned models—trained via methods such as Direct Preference Optimization (DPO), Proximal Policy Optimization (PPO), and Group Relative Policy Optimization (GRPO)—are increasingly the standard. These methods have been highly successful at aligning models with human preferences, yet their impact on output diversity remains

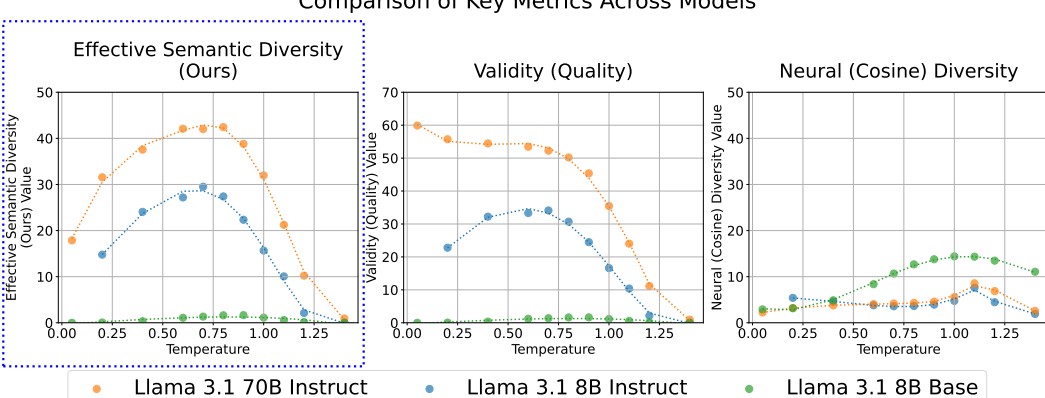

Figure 1: Diversity and Quality metrics on our open-ended programming dataset when modulating the Temperature parameter across different models. We use the CODE-BERTSCORE (Zhou et al., 2023) model for neural cosine diversity.

disputed. The dominant research narrative is that preference tuning harms diversity, and many open questions persist, including the effect of model size and the ability to generate unique content.

We introduce a principled framework for measuring high-quality diversity that:

- Requires no human evaluation at inference time
- Accounts for the fundamental quality-diversity interplay
- Enables meaningful comparison across model families and training techniques

In Figure 1 we illustrate our implementation of effective semantic diversity for three LLAMA-3.1 family models under varying temperature settings. Using neural cosine diversity alone, the base LLM appears most diverse, despite producing an extremely low proportion of valid (high-quality) generations. Our effective semantic diversity metric penalizes both excessively low and excessively high temperatures, as well as models that struggle to generate coherent content.

With this framework, our experiments reveal insights into preference-tuning and the diversity-quality trade-off. Using diversity metrics that do not explicitly consider quality, preference tuning may indeed reduce diversity relative to supervised fine-tuning (SFT). However, in both code generation and creative writing, we find that preference-tuned models exhibit greater effective semantic diversity. A deeper analysis in code generation shows that when restricting to high-quality generations only, preference-tuned models produce less semantically diverse content within that subset. Nonetheless, preference tuning increases the proportion of high-quality outputs to such an extent that this gain outweighs the reduction in diversity per high-quality sample.

In code generation, preference tuning—especially reinforcement learning (RL)—is associated with reduced lexical and syntactic diversity but no loss in semantic diversity. For open-ended creative writing, preference tuning is linked to greater diversity in lexical patterns. Finally, when evaluating parameter efficiency for generating unique programs within a fixed sampling budget—for example, when creating unique synthetic data—we find that smaller models, down to around 500 million parameters, are often the most efficient choice.

## 2  Background and Related Work

**LLM Alignment with Preference Tuning**. As LMs (Bengio et al., 2000; Radford et al., 2019) have become more powerful, work has been done to improve their instruction-following ability and to mitigate the likelihood of generating undesirable content. RLHF with PPO (Schulman et al., 2017; Ziegler et al., 2019; Ouyang et al., 2022) has emerged as a highly effective technique for aligning LLMs with human preferences. Subsequently,

numerous alternative methods to PPO have been proposed, such as DPO (Rafailov et al., 2024), Rejection Sampling (Touvron et al., 2023), and GRPO (Shao et al., 2024). In numerous works, it has been reported that over-optimizing for the reward model eventually leads to incoherent or undesirable outputs (Ziegler et al., 2019; Stiennon et al., 2020; Korbak et al., 2022). It is suggested that an optimization strategy that places all probability on the highest-reward outcome is optimal for this loss function but will inevitably lead to distribution collapse (Lanchantin et al., 2025). In preference-tuning LLMs, KL-divergence regularization has been instrumental in mitigating this effect, as it allows the preference-tuned model to retain some of the distributional properties of the base LLM (Korbak et al., 2022). Therefore, in theory, KL-regularized RL should preserve some attributes of diversity present in base models.

**Approaches to Measuring Diversity.** Due to the high cost of human evaluation, popular methods for automatically measuring diversity typically fall into either lexical or neural approaches. In the earlier days, when LMs often struggled to generate rich content, lexical metrics were commonly used to assess diversity. These metrics generally involve calculating summary statistics over $n$-grams, such as **Distinct-N** (Li et al., 2016; Du & Black, 2019) and **Self-BLEU** (a modified BLEU metric) (Zhu et al., 2018), and remain widely used today (Guo et al., 2024; Shaib et al., 2024). As early foundation models for language representation, such as BERT (Devlin et al., 2019), gained popularity and were adopted for modeling sentence similarity (Zhang et al., 2019; Reimers & Gurevych, 2019) and code similarity (Feng et al., 2020; Zhou et al., 2023), they were eventually proposed for measuring diversity in natural language (Lai et al., 2020; Tevet & Berant, 2021; Stasaski & Hearst, 2022). However, they have not yet been applied to measuring diversity in programming languages.

While lexical and neural diversity metrics have been used to evaluate whether one LLM is more diverse than another (Kirk et al., 2023; Guo et al., 2024), to our knowledge no prior work has assessed whether neural diversity metrics truly capture the diversity of effective semantic content in LLM-generated outputs. Given that LMs range from small models that often produce incoherent content to large, powerful models with sophisticated capabilities, varying safety attributes, and distinctive "styles," it remains unclear whether existing diversity metrics can robustly reflect meaningful semantic content across such a wide range of distributions. Models used for evaluating diversity, such as Sentence-BERT (Reimers & Gurevych, 2019), are typically trained and evaluated on human-authored text; for diversity assessment, however, they are expected to generalize to highly diverse, potentially off-distribution sets of LLM-generated text. The closest prior attempt to evaluate lexical and neural diversity metrics is Tevet & Berant (2021), which examined whether neural diversity metrics can capture variation induced by the sampling temperature parameter and diversity in human-written content; in the latter case, neural models were found to underperform human judgment.

**Diversity of LLM Content**. Given the expense of human evaluation, most insights into the novelty and diversity of LLM outputs are based on linguistic and neural measures of diversity. Zhang et al. (2020) introduce the notion of a quality-diversity trade-off in natural language generation when evaluating sampling algorithms such as nucleus sampling (Holtzman et al., 2019) and temperature sampling. McCoy et al. (2023) investigate whether smaller LMs exhibit linguistic novelty by evaluating combinations of $n$-grams absent from the training corpus. With the advent of more chat-based LLMs like CHATGPT (Ouyang et al., 2022; OpenAI et al., 2023), expectations for these models to serve as creative assistants and produce diverse responses have risen, but their capabilities have been questioned. For example, the first release of CHATGPT was reported to be incapable of generating diverse jokes (Jentzsch & Kersting, 2023). In a more recent study, Kirk et al. (2023) fine-tune 7B ALPACAFARM (Dubois et al., 2024) models with SFT and PPO for neural text summarization and find that PPO reduces both lexical and neural diversity in the summaries. Padmakumar & He (2023) show that human-written essays assisted by an RLHF-tuned model are less diverse than those assisted by a base model when assessed using neural diversity measures. Guo et al. (2024) benchmark the lexical, syntactic, and neural diversity of 7B parameter language models but do not probe whether preference-tuned models are more or less diverse than SFT or base alternatives. A separate line of work has examined LLMs from a social cognition perspective, investigating whether LLMs reflect the opinions and conceptual

associations of macro-level human populations (Santurkar et al., 2023; Murthy et al., 2024). Recent work has also explored the role of diversity metrics in evaluating and improving synthetic data quality for LLMs: Yu et al. (2023) investigate LLMs as attributed training data generators; Divekar & Durrett (2024) propose SYNTHESIZRR, a retrieval-augmented framework for generating synthetic datasets with greater lexical and semantic diversity; Chen et al. (2024) study the downstream impact of synthetic data diversity on training LLMs; and Miranda et al. (2025) introduce a quantitative metric, the diversity coefficient, to measure variability in natural language datasets. While these works find that LLMs may not reflect the diversity of *entire human populations*, it remains unclear how this impacts their ability to generate effectively diverse content—particularly when faced with competing constraints of quality and helpfulness.

## 3 Measuring Effective Semantic Diversity

### 3.1 Problem Formulation

Let $\mathcal{D} = \{x_i\}_{i=1}^N$ denote a dataset of prompts, where each prompt $x_i$ is designed to elicit a range of possible outputs from a language model. For each prompt $x_i$, we generate $K$ outputs $\mathcal{P}_i = \{g_i^1, g_i^2, \dots, g_i^K\} \sim f(\cdot \mid x_i)$, where $f$ denotes the generative distribution of the language model under evaluation.

To evaluate the overall diversity of a model on the dataset $\mathcal{D}$, we compute the average diversity score:

$$\text{AvgDiv}_m = \frac{1}{N} \sum_{i=1}^N \text{Div}_m(\mathcal{P}_i), \tag{1}$$

where $\text{Div}_m$ is a diversity metric and $m$ denotes the specific type of metric being used. In our formulation below, $\text{Div}_m$ refers to our proposed effective semantic diversity metric. However, $\text{Div}_m$ could represent any diversity metric, including lexical metrics like Distinct $n$-grams and average cosine distance.

**Validity function.** To determine diversity at the individual prompt level, we define a validity function $V : \mathcal{G} \to [0,1]$, where $\mathcal{G}$ is the space of generations.

**Semantic funtion.** Crucially, we determine the semantic uniqueness of generations based on their meaning rather than superficial textual differences. Two outputs may differ significantly in their lexical form yet convey identical semantic content—a distinction that purely lexical or syntactic metrics fail to capture. Our goal is to define a diversity metric over a set of generations that captures both this semantic distinctiveness and the validity of outputs.

We define a semantic function $S : \mathcal{G} \to \mathcal{S}$, where $\mathcal{S}$ is the semantic space. Two generations $g, g' \in \mathcal{G}$ are said to be **semantically equivalent** if and only if $S(g) = S(g')$. The detailed definition of $S$ on each domain is described in Section 3.2.

We then formalize effective semantic diversity in two complementary ways:

**Semantic diversity.** We define the **effective semantic diversity** of a generation set $\mathcal{P}_i$ as the number of semantically unique valid generations, normalized by the total number of generations in the set:

$$\text{Div}_{\text{fixed}}(\mathcal{P}_i) = \frac{|\text{Set}(\{S(g_i^k) \mid g_i^k \in \mathcal{P}_i, \ V(g_i^k) = 1\})|}{K}, \tag{2}$$

where $\text{Set}(\cdot)$ denotes the set of unique elements and $|\cdot|$ denotes cardinality. This measures the proportion of generations that are both valid/high-quality and semantically unique within the group of generations.

However, we found that Equation (2) can be confounded by the number of samples under consideration if analyzing a variable-sized subset of valid-only programs; see Appendix A.2 for more detail. To address these concerns, we adopt the pairwise diversity metric suggested in Tevet & Berant (2021):

> **Input Description:**
> - Multiple datasets.
> - Each dataset consists of four real numbers: a, b, c, d.
> - There are no more than 30 datasets.
>
> **Example Input:**
> ```
> 35.68 139.77 51.15 359.82
> 01.37 103.92 41.78 272.25
> 51.15 359.82 -34.58 301.52
> ```
>
> **Function Signature:**
> Write a function f(inputs) that processes the list of tuples where each tuple contains four real numbers.
>
> ```python
> from typing import List, Tuple
> def f(inputs: List[Tuple[float, float, float, float]]):
>     '''
>     inputs: a list of tuples, where each tuple contains four real numbers
>     '''
> ```

Figure 2: An example of an open-ended problem description from our dataset.

$$\text{Div}_{\text{pair}}(\mathcal{P}_i) = \frac{1}{\binom{K}{2}} \sum_{\substack{g_i^j, g_i^k \in \mathcal{P}_i, \\ 1 \leq j < k \leq K}} d_{\text{sem}}(g_i^j, g_i^k), \tag{3}$$

where the effective semantic distance function $d_{\text{sem}} : \mathcal{G} \times \mathcal{G} \to \{0, 1\}$ is defined as:

$$d_{\text{sem}}(g_i^j, g_i^k) = \begin{cases} 0 & \text{if } V(g_i^j) = V(g_i^k) = 0, \\ 0 & \text{if } V(g_i^j) = V(g_i^k) = 1 \text{ and } S(g_i^j) = S(g_i^k), \\ 1 & \text{otherwise.} \end{cases} \tag{4}$$

This pairwise approach normalizes by the total number of possible pairs, making it robust to variations in the number of valid generations across different prompts. However, in domains such as natural language where strict notions of semantics do not apply, we define a soft approach to measuring $d_{\text{sem}}$ as:

$$d_{\text{sem}}(g_i^j, g_i^k) = V(g_i^j) \times V(g_i^k) \times (1 - Sim(g_i^j, g_i^k)) \tag{5}$$

Here $Sim : \mathcal{G} \times \mathcal{G} \to [0, 1]$ is a similarity function that does not explicitly consider quality, for example, assigned by a human or AI judge.

## 3.2 Dataset, Validity, and Semantic Equivalence Checking for Programs

We chose to instantiate methods of validity and semantic equivalence checking using both programs and natural language. We found it advantageous to use programs to avoid the scenario of using LLMs to evaluate LLMs for diversity, especially if we seek to evaluate increasingly powerful LLMs with weaker LLMs. Furthermore, there has been relatively little empirical work that studies the robustness of using LLMs to judge the diversity of LLM-generated content. Lastly, our implementation of effective semantic diversity for programs avoided the costs of utilizing commercial LLMs as a judge. Research in programming language semantics has a rich history in the formalization of program correctness, termination, and semantic equivalence (Hoare, 1969; Plotkin, 1981; Floyd, 1993; Pierce, 2002). Using program execution on test cases, we can confirm two programs are semantically distinct if they compute different values over the same suite of input test cases[1]. Additionally, we can

---

[1] *N.B.* Test cases may fail to test edge cases where two programs may differ. However, instead of over-reporting correctness, this phenomenon would only *penalize* diversity when two programs are equivalent on most test cases and hypothetically differ on some edge-case, which we find tolerable.

impose constraints of validity or quality that are easy to check: such as the program always returning a value without syntax or other runtime errors.

The key question, however, is to define open-ended programming tasks. Generating programs as an open-ended task is not necessarily radical: in chat applications, users may desire to brainstorm programs or code projects. Moreover, generating synthetic programming tasks is already an important part of improving foundation LLMs for code (Dubey et al., 2024; Shypula et al., 2024).

We constructed our dataset by adapting competitive programming-style problems into open-ended abstracted programming tasks. In Figure 2, we show an example problem description from our dataset. For each problem description in our dataset, we provide an "Input Description" in natural language specifying the input format, an "Example Input" demonstrating potential inputs the function would handle, and a "Function Signature" providing a concrete specification of the function name and typing hints for the inputs. We manually reviewed, and if necessary, edited all final descriptions, removing any potential references to the original programming task, standardizing the function name to $f(...)$, and using highly generic argument names. We used competitive programming problems from CODENET (Puri et al., 2021) and accompanying test cases from ALPHACODE (Li et al., 2022) as a starting point for our dataset. In total we have 108 unique programming tasks/prompts that were adapted from CODENET into this open-ended format. Using each of these, we can sample an arbitrary number of generations to calculate $\text{Div}_m(\mathcal{P}_i)$. Further details about the test set, its creation, and our execution environment are provided in Appendix A.3.

### 3.3 Dataset for Natural Language

For natural language, we created our dataset by taking a subsample of 100 creative writing prompts from the WRITINGPROMPTS dataset (Fan et al., 2018). We took a random subset of the test set, prioritized [WP] tagged prompts, and avoided prompts that could generate sensitive or inappropriate stories.

## 4 Experimental Setup

**Diversity Metrics.** We now formalize the diversity metric $\text{Div}_m(\mathcal{P}_i)$ with particular focus on our proposed notion of **effective semantic diversity.** As introduced in Section 3.1, this metric relies on the choice of valid function $V$ and semantic function $S$.

Let $\mathcal{T} = \{t_1, \ldots, t_L\}$ be a fixed set of test inputs. For any generated program $g$, let $g(t_l)$ denotes its execution on test case $t_l \in \mathcal{T}$, and $o_l$ denotes the corresponding output. Specifically, for code we define the **validity function** $V : \mathcal{G} \to \{0, 1\}$ as follows:

$$V(g) = \begin{cases} 1 & \text{if } g(t_l) \text{ executes without error and produces non-null output } \forall l \in \{1, \ldots, L\}, \\ 0 & \text{otherwise.} \end{cases}$$

In other words, a program $g$ is considered valid if it runs without raising any errors (e.g., SyntaxError, ValueError) and produces non-null outputs for all test cases in $\mathcal{T}$.

In the domain of natural language, we prompt a LLM judge with a set of criteria to judge the quality of the creative generation and then normalize the score to fall between $[0, 1]$.

Next, we define the **semantic function** $S : \mathcal{G} \to \mathcal{O}^L$, where $\mathcal{O}$ denotes the output space. The function maps a program $g$ to its output trace on the test set:

$$S(g) = (g(t_1), g(t_2), \ldots, g(t_L)).$$

Two programs $g$ and $g'$ are considered **semantically equivalent** if and only if their outputs are identical across all test cases:

$$S(g) = S(g') \iff g(t_l) = g'(t_l) \quad \forall l \in \{1, \ldots, L\}.$$

For natural language, for *Sim*, the semantic similarity function that does not explicitly consider quality, we prompt an LLM judge to evaluate the conceptual and thematic overlap

| Model Family | Base | SFT | DPO | RL |
|---|---|---|---|---|
| **LLaMA 2 7B** | Llama-2-7b-hf | tulu-2-7b | tulu-2-dpo-7b | Llama-2-7b-chat-hf (PPO) |
| **LLaMA 2 70B** | Llama-2-70b-hf | tulu-2-70b | tulu-2-dpo-70b | Llama-2-70b-chat-hf (PPO) |
| **LLaMA 3.1 8B** | Llama-3.1-8B | Tulu-3-8B-SFT | Tulu-3-8B-DPO Llama-3.1-8B-Instruct | Tulu-3-8B (PPO) Tulu-3.1-8B (GRPO) |
| **LLaMA 3.1 70B** | Llama-3.1-70B | Tulu-3-70B-SFT | Tulu-3-70B-DPO Llama-3.1-70B-Instruct | Tulu-3-70B (PPO) |

Table 1: **Model Post-Training Categorization.** We organize all models under consideration by their base model and post-training method. All models to the right of "Base" are post-trained with the specified method from the base model.

between two generations and normalize the score. For all LLM judge tasks we utilize `gpt-4.1-mini-2025-04-14`. Due to the scale of our experiments, we subsampled 32 pairs with replacement from all possible pairs for each set of generations.

For the remaining diversity metrics, we adopt Equation (3) and utilize the following distance functions in place of $d_{\text{sem}}(g, g')$. For **Lexical diversity,** we use Expectation-Adjusted Distinct $n$-grams (**EAD**) with $n$-grams of length $n=4$. For **syntactic diversity** in code, we adapt the Distinct-N metric to the Abstract Syntax Tree (**AST**) of each generated program: to isolate the syntactic structure of a program (e.g., for-loop instead of recursion) from superficial choices (e.g., variable names), we canonicalize all program identifiers. For two programs, we calculate the ratio of the number of unique subtrees of height $H$ across both programs to the total number of subtrees of height $H$ in both programs, where $H=4$. In order to measure **neural diversity** for code, we adapt existing methods of neural diversity metrics (Tevet & Berant, 2021) to our domain using CODEBERTSCORE (Zhou et al., 2023). We use CODEBERTSCORE because it closely resembles the models used in the NLP literature to evaluate diversity (Tevet & Berant, 2021) and has been evaluated as effective in the evaluation of neural codes. We also attempted to use CODELLAMA-7B-INSTRUCT embeddings and ICESCORE (Zhuo, 2024) to evaluate diversity and found CODELLAMA similar to CODEBERTSCORE and ICESCORE to correlate strongly with temperature, such as the lexical diversity metric. See the the additional results in Appendix A.1. For natural language, we report the average score from the LLM judge $\mathcal{D}$ before normalizing it by the generation's quality.

**Models and Sampling.** The main empirical questions we seek to answer in our work surround the effects of different post-training algorithms and model size on diversity (SFT, DPO, PPO, GRPO, etc). In order to isolate the effects of different post-training strategies, we utilize the TULU-2 (Ivison et al., 2023) and LLAMA-2 (Touvron et al., 2023) as well as the TULU-3 (Lambert et al., 2024) and LLAMA3.1 (Dubey et al., 2024) families of models. TULU-2 consists of post-trained LLAMA-2 models, and TULU-3 consists of post-trained LLAMA3.1 models. In Table 1, we provide a categorization of the post-training methods. Using these we can compare a SFT-tuned model to a DPO-tuned model from the same family. For each problem description $x_i$ in our dataset, we generate $K = 32$ programs, yielding 3,456 total programs sampled for each experiment. In cases where models are post-trained with numerous algorithms, we often categorize the model with the most aggressive algorithm used, e.g. we would categorize a model adapted with DPO and PPO as PPO.

In addition to our questions regarding the effects of post-training and model size, we also investigate which sizes and classes of models are most *efficient* on a *per-parameter basis* in generating unique examples. This is an attempt to understand which types of models could be optimal for large-scale synthetic data generation. We relax the strict necessity to pair models by the same base model, and use models from QWEN2.5-CODER (Hui et al., 2024), QWEN2.5 (Yang et al., 2024), and as well as DEEPSEEK-R1-DISTILL (Guo et al., 2025) family. Our goal was to test a broader set of sizes as well as models tuned specifically for reasoning tasks (i.e. DEEPSEEK-R1).

| Comparison | Validity | | Semantic | | Syntactic | | Lexical | | Neural | |
|---|---|---|---|---|---|---|---|---|---|---|
| | W (p) | ES (d) | W (p) | ES (d) | W (p) | ES (d) | W (p) | ES (d) | W (p) | ES (d) |
| BASE VS. INST (ALL) | **<0.001** | **1.33** | **<0.001** | **1.34** | **<0.001** | **-1.10** | **0.028** | -0.47 | **<0.001** | **-1.53** |
| BASE VS. INST-SFT | **<0.001** | **1.37** | **<0.001** | **1.36** | 0.266 | -0.34 | 0.339 | 0.21 | 0.339 | -0.42 |
| BASE VS. INST-DPO | **0.006** | **1.11** | **0.005** | **1.10** | **0.014** | **-0.84** | **<0.001** | **-1.68** | **<0.001** | **-2.37** |
| BASE VS. INST-RL | **<0.001** | **1.67** | **<0.001** | **1.54** | **0.012** | **-1.20** | **0.035** | -0.66 | **<0.001** | **-2.12** |
| BASE VS. INST-PREF | **<0.001** | **1.34** | **<0.001** | **1.29** | **<0.001** | **-1.45** | **0.001** | -0.77 | **<0.001** | **-2.29** |
| SFT VS. DPO | 0.151 | 0.51 | 0.266 | 0.39 | **<0.001** | **-0.90** | **0.003** | -0.63 | **<0.001** | **-1.52** |
| SFT VS. RL | **0.004** | **3.19** | **0.004** | **2.49** | **0.004** | **-2.43** | **0.004** | **-1.20** | **0.004** | **-2.29** |
| SFT VS. PREF | **<0.001** | **0.97** | **0.002** | 0.77 | **<0.001** | **-1.30** | **<0.001** | **-0.83** | **<0.001** | **-1.86** |
| DPO VS. RL | 1.0 | 0.10 | 0.301 | -0.14 | **0.039** | -0.60 | **0.004** | **-0.78** | 1.0 | -0.01 |
| SM. VS. LG. | **0.030** | 0.20 | **0.017** | 0.26 | 0.170 | 0.15 | 0.485 | 0.09 | 0.269 | 0.12 |
| SM. VS. LG. BASE | 0.063 | 0.57 | 0.063 | 0.56 | 0.063 | 0.74 | 0.063 | 0.69 | **0.031** | **0.80** |
| SM. VS. LG. INST | 0.111 | 0.20 | 0.070 | 0.27 | 0.683 | 0.06 | 0.539 | -0.11 | 0.759 | -0.05 |
| SM. VS. LG. PREF | 0.330 | 0.13 | 0.277 | 0.20 | 0.454 | 0.10 | 0.330 | -0.19 | 0.389 | 0.14 |

Table 2: **Model Comparison Results.** Results from Wilcoxon's Signed-Rank Test p-values: **W (p)**, and Effect Size measured by Cohen's D: **ES (d)**. Bold p-values are below 0.05, and bold d-values have an absolute value greater than 0.8 (large effect size). For effect sizes, positive values indicate the second model type in the comparison has higher values. "Inst" = Instruction-tuned (Post-Trained), "Pref" = Preference-tuned and combines RL and DPO models. The columns correspond to metrics in Section 4 (e.g. "Semantic" corresponds to effective semantic diversity).

**Prompt selection.** The choice of prompt can affect the nature of generations. Because of this, we created three separate prompt templates: a zero-shot prompt, a two-shot prompt, and a two-shot prompt with chain-of-thought reasoning. This design allows us to probe generation behavior across a variety of settings. The few-shot examples included in the prompts were simple, manually written examples shared across all problems in the dataset. We provide the examples in Appendix A.6.

**Statistical tests.** We aim to determine if a factor such as post-training algorithm or model size increases or decreases diversity compared to another. To rigorously summarize these results, we report the two-tailed Wilcoxon Signed-Rank Test (to measure significance) and Cohen's D (to measure effect size) for $AvgDiv_m$ over the entire dataset for each statistic. A benefit of the non-parametric Wilcoxon statistical test is that we can make rigorous conclusions even if only limited samples are available. We always pair models from the same family and vary whether the model is larger fine-tuned with a different strategy *while fixing all other factors* unless otherwise noted. For example, when isolating model size, we would compare LLAMA3.1-8B-INSTRUCT with Zero-Shot prompting to LLAMA3.1-70B-INSTRUCT with Zero-Shot prompting, and so on. Because of the differences in post-training pipelines and to avoid confounding factors, we avoid pairing TULU post-trained models with LLAMA preference-tuned models.

# 5 Experimental Results

## 5.1 Effect of Post-Training and Preference-Tuning on Diversity

**Higher effecitve semantic diversity introduced by preference tuning.** We summarize results across all diversity metrics when comparing base models and their instruction-tuned counterparts, as well as when comparing different post-training techniques, for programming tasks in Table 2 and natural language tasks in Table 3. Each row compares two fine-tuning approaches; for each metric, the columns report the statistical significance **W (p)** and effect size **ES (d)**. A statistically significant positive effect indicates that the second post-training strategy outperforms the first for that metric.

| Comparison | Validity | | Semantic | | Lexical | | Neural | |
|---|---|---|---|---|---|---|---|---|
| | W (p) | ES (d) | W (p) | ES (d) | W (p) | ES (d) | W (p) | ES (d) |
| BASE VS. INST (ALL) | **<0.001** | **2.02** | **<0.001** | **1.50** | **<0.001** | **1.65** | **<0.001** | **-3.60** |
| BASE VS. INST-SFT | **<0.001** | **2.19** | **0.001** | **1.66** | **<0.001** | **2.53** | **<0.001** | **-2.61** |
| BASE VS. INST-DPO | **0.001** | **1.51** | **0.014** | **0.89** | **<0.001** | **2.27** | **<0.001** | **-4.27** |
| BASE VS. INST-RL | **<0.001** | **4.88** | **<0.001** | **3.91** | **0.022** | 0.73 | **<0.001** | **-5.68** |
| BASE VS. INST-PREF | **<0.001** | **2.18** | **<0.001** | **1.53** | **<0.001** | **1.34** | **<0.001** | **-4.58** |
| SFT VS. DPO | **<0.001** | **1.52** | **0.001** | **0.96** | **<0.001** | **1.12** | **<0.001** | **-1.61** |
| SFT VS. RL | **0.004** | **5.79** | **0.008** | **2.76** | **0.004** | **3.60** | **0.004** | **-4.67** |
| SFT VS. PREF | **<0.001** | **2.36** | **<0.001** | **1.54** | **<0.001** | **1.48** | **<0.001** | **-2.39** |
| DPO VS. RL | 0.055 | 0.51 | **0.025** | 0.59 | 0.82 | 0.28 | **0.02** | -0.40 |
| SM. VS. LG. | **<0.001** | 0.41 | **<0.001** | 0.53 | **0.04** | -0.21 | **0.009** | -0.20 |
| SM. VS. LG. BASE | **0.031** | **3.04** | **0.031** | **3.35** | 0.844 | 0.05 | 0.062 | **-1.21** |
| SM. VS. LG. INST | **0.001** | 0.40 | **0.006** | 0.42 | **0.003** | -0.32 | 0.055 | -0.26 |

Table 3: **Creative Writing (Natural Language) Model Comparison Results.** Bold p-values are below 0.05, and bold d-values have an absolute value greater than 0.8 (large effect size)

| Comparison | Semantic | | Syntactic | | Lexical | | Neural | |
|---|---|---|---|---|---|---|---|---|
| | W (p) | ES (d) | W (p) | ES (d) | W (p) | ES (d) | W (p) | ES (d) |
| SFT VS. DPO | 0.052 | -0.71 | 0.791 | -0.35 | 1.000 | -0.12 | **0.016** | **-0.89** |
| SFT VS. RL | **0.004** | **-3.12** | 0.426 | -0.28 | 0.820 | 0.08 | **0.004** | **-2.25** |
| SFT VS. PREF | **<0.001** | **-1.31** | 0.473 | -0.32 | 0.946 | -0.05 | **<0.001** | **-1.39** |
| DPO VS. RL | 0.301 | -0.43 | **0.012** | **-1.16** | **0.008** | **-1.24** | 0.570 | -0.05 |

Table 4: **Validity-Controlled Model Comparison Results for Code.** Bold p-values are below 0.05, and bold d-values have an absolute value greater than 0.8 (large effect size)

We observe a clear trend: all post-training techniques increase both effective semantic diversity and validity relative to base models. RL methods, in particular, yield substantial improvements over SFT in effective semantic diversity. We also find an interesting pattern: preference tuning tends to substantially reduce syntactic and lexical diversity in programming tasks, yet increases these metrics in natural language creative writing. These results suggest that in domains requiring high-quality and diverse outputs, preference-tuned models can outperform both SFT and base models. Furthermore, in creative writing—where diverse word choice and stylistic variety are often desirable—preference-tuned models may hold an advantage in stylistic capabilities. Additional experiments are provided in Appendix A.7 and Appendix A.8.

**Semantic diversity driven by higher quality.** In Table 4, we conduct statistical tests similar to those in Table 2 for code, but restrict the analysis to only valid generations. We find that within this subset, preference-tuned models generally have more semantic duplicates than their SFT-tuned counterparts. This suggests that while preference tuning, particularly RL, tends to increase semantic duplication within high-quality generations, the overall increase in the number of high-quality generations more than compensates for this effect.

## 5.2 Effects of Model Size

**Larger models increase semantic diversity.** In Table 2 and Table 3, we compare small and large models within the same family. In both code and natural language, larger models generally exhibit higher semantic diversity. In code, this increase does not come at the expense of lexical or syntactic diversity; however, in natural language, we observe a small but significant decrease in lexical diversity. We attribute the improvements in quality and effective semantic diversity to the well-documented strengths of larger models in helpfulness and quality.

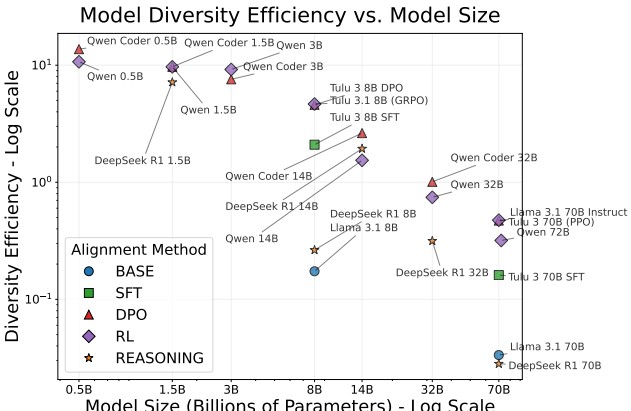

Figure 3: **Model efficiency**: We plot the parameter efficiency of a model in generating unique examples vs. model size (log-scale) on our programming dataset.

**Smaller models may be more compute-efficient for generating unique examples.** While larger models tend to produce more diverse outputs per generation, we also examine which models are most efficient given a fixed computational budget. Consider a researcher aiming to generate a large number of unique synthetic programs: is it better to take fewer samples from a very large LLM, many samples from a smaller LM, or find a "sweet spot" in between? In Figure 3, we explore this question by plotting the parameter efficiency for diversity against model size, where efficiency is measured as the effective semantic diversity (per Equation (2)) normalized by the number of model parameters. For a fixed sampling budget of 32 generations per prompt in the programming dataset, smaller models are consistently more efficient *on a per-parameter basis*. We believe that further work should be done to investigate whether this pattern continues to hold as the number of samples increases, to ensure semantically unique programs do not saturate faster for certain model classes than others.

## 6   Discussion and Conclusion

We propose a novel strategy for studying effective semantic diversity by leveraging code execution to jointly measure and balance quality and diversity. In the natural language domain, we use LLM judges as a proxy for human evaluation. Using these methodologies, we conduct an extensive empirical analysis of LLM diversity and find counterintuitive insights into how factors such as post-training and model size meaningfully influence diversity. We encourage further work to further broaden the range of empirical questions explored in this area.

**Acknowledgements**

We would like to thank Christopher Watson, Cassidy Laidlaw, and other members of Berkely AI Research for feedback on our draft. This material is partly based on research sponsored in part by the Air Force Research Laboratory (agreement number FA8750-19-2-0200 and award W911NF-20-1-0080), an Amazon Research Award Fall 2023, and an Amazon/ASSET Gift for Research in Trustworthy AI. The U.S. Govt. is authorized to reproduce and distribute reprints for Governmental purposes notwithstanding any copyright notation thereon. The views and conclusions contained herein are those of the authors and should not be interpreted as necessarily representing the official policies or endorsements, either expressed or implied, of the Air Force Research Laboratory or the U.S. Government.

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

# A  Appendix

## A.1  Additional Neural Diversity Metrics

In addition to using CODEBERTSCORE, we also demonstrate neural diversity with respect to temperature sweeps similar to those in Figure 1 using CODELLAMA-7B-INSTRUCT (Roziere et al., 2023) embeddings. We also use an LLM-as-a-judge to calculate pairwise similarity using ICESCORE (Zhuo, 2024) using gpt-4o-mini as the LLM-judge. We find that ICESCORE correlates heavily with temperature and does not seem to reflect quality. Similarly with CODELLAMA embeddings, the LLAMA8B-BASE model's generations are consistently considered more diverse than the INSTRUCT variants despite the base model generally producing far more low-quality and invalid outputs.

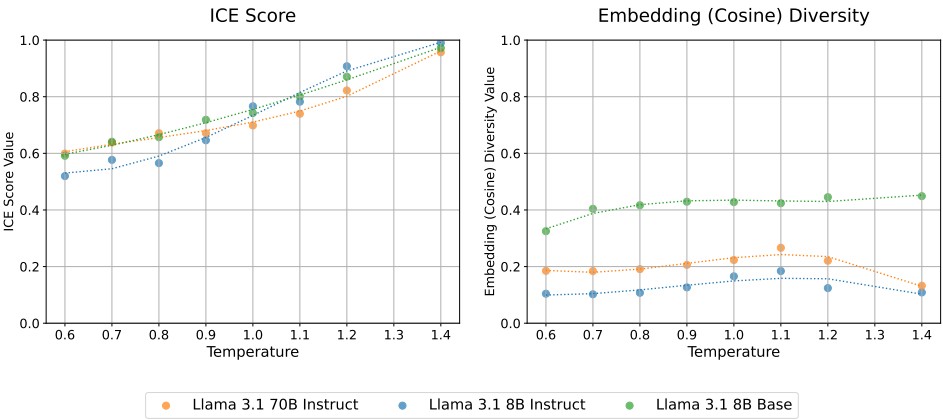

Figure 4: Neural diversity metrics when modulating the Temperature parameter across different models. We report neural diversity metrics using the ICESCORE (Zhuo, 2024) and cosine diversity of CODELLAMA-7B-INSTRUCT embeddings.

## A.2  Sample Size Confounding Diversity and Analysis of Pairwise Diversity Metric

In practice, we found that varying the size of the subset under consideration could dramatically affect the calculation of Equation (2): for the same model, as samples tended to get larger, this metric would decrease. When we use this metric for most of our experiments, it is sufficient, because we keep the number of samples constant. However, when we analyze the subset of valid-only programs, the size of this population may vary from model to model depending on the average quality of generations. Thus it is necessary to use Equation (3).

**Analysis of Pairwise Diversity Metric.** We provide an analysis of Equation (3). For a given large language model $f$, we assume that only a finite number $K$ of distinct semantic meanings can be generated by $f$. We first establish that the original semantic diversity metric converges to zero as the number of sampled responses tends to infinity. Specifically, the original semantic diversity metric is defined as

$$\frac{N}{n},$$

where $N$ is the number of distinct semantic clusters, and $n$ is the number of sampled responses. Since only a finite number of semantic meanings can be generated by $f$, the number of semantic clusters $N$ is bounded above, implying the existence of a constant $C_1 > 0$ such that $N \geq C_1$. Therefore, we have

$$\frac{N}{n} \geq \frac{C_1}{n} \quad \text{for all sufficiently large } n.$$

Now, observe that

$$\lim_{n\to\infty} \frac{C_1}{n} = 0,$$

so applying the squeeze theorem, we conclude that

$$\lim_{n\to\infty} \frac{N}{n} = 0.$$

Next, we show that the new metric defined in Equation (3), converges to a constant as $n \to \infty$. As before, we assume that there are $K$ distinct semantic meanings in total, and let $\pi_k$ denote the proportion of responses corresponding to the $k$-th semantic meaning. This implies that the number of times each semantic meaning is sampled is $\pi_k n$, where $\sum_{k=1}^{K} \pi_k = 1$. Thus, we have

$$\sum_{p_i, p_j \in P, i>j} m_{\text{dist}}(p_i, p_j) = \sum_{k\neq h} (\pi_k n) \cdot (\pi_h n) = \sum_{k\neq h} (\pi_k \pi_h) n^2,$$

where the summation is taken over distinct semantic meanings $k$ and $h$, and $m_{\text{dist}}(p_i, p_j)$ measures the semantic distance between generations. Moreover, the number of possible response pairs is

$$\binom{|P|}{2} = \frac{n(n-1)}{2}.$$

Thus, the new metric becomes

$$\frac{1}{\binom{|P|}{2}} \sum_{p_i, p_j \in P, i>j} m_{\text{dist}}(p_i, p_j) = \frac{2\sum_{k\neq h}(\pi_k \pi_h) n^2}{n(n-1)} = 2\sum_{k\neq h}(\pi_k \pi_h) + \frac{2\sum_{k\neq h}(\pi_k \pi_h)}{n-1}.$$

Finally, we have

$$\lim_{n\to\infty} \left\{ 2\sum_{k\neq h}(\pi_k \pi_h) + \frac{2\sum_{k\neq h}(\pi_k \pi_h)}{n-1} \right\} = 2\sum_{k\neq h} \pi_k \pi_h.$$

That is, as $n \to \infty$, the value of the new metric converges to the constant value:

$$2\sum_{k\neq h} \pi_k \pi_h.$$

### A.3 Dataset Creation and Additional Details

We created our dataset through a multi-step process starting from the CODENET dataset and testcases from ALPHACODE. The process involved problem standardization, language model assistance for scaling, and manual validation.

**Initial Processing.** We began by randomly selecting a single problem description from IBM CodeNet. We used this to as a seed to construct examples for a 1-shot prompt for a Language Model to assist us. Specifically we manually wrote one example of the following outputs that should be generated for each individual problem description from CODENET

1. A canonicalized problem description

2. A wrapper function that would take any generation for that function, parse inputs for the function, and instrument the generation with the entire suite of test cases

3. A property-based testing function for generating additional test cases when needed

**Dataset Expansion.** We then wrote prompt templates for both `gpt-3.5-turbo-0125` and `gpt-4o-2024-11-20` (depending on the iteration). These templates were designed to take our 1-shot prompt, and then prompt the LLM to repeat the same for the new example we select from CODENET. We randomly sampled over 300 programs from CodeNet and attempted to generate the three components for each original problem description from CODENET. We

then saved these into individual files, and also wrote them into an HTML document for manual review.

**Manual Review and Selection.** For the over 300 problem ids that were processed, we then manually inspected the output components checking for the following criteria:

1. The language model correctly formatted inputs into our desired format
2. The problem appeared novel relative to the existing problems we had already added to our dataset.

We tracked the original CodeNet problem IDs and validation results in a spreadsheet. These were the 108 examples then used for our dataset.

**Manual Editing of Problem Descriptions.** After selecting our problems, we then manually went through each of the three components, and manually edited them to be correct: a large amount required revisions, as the LLM assistant made mistakes. We saved our manually edited components to be further processed.

**Test Case Integration.** For each of the individual problem descriptions, we then merged test cases from CODENET and ALPHACODE. We required at least 10 test cases per problem. For the three problems that lacked sufficient test cases, we used our property-based testing scripts to generate 100 additional cases, such that we would have sufficient coverage. We then saved the canonicalized problem descriptions, the function to extract and parse input test cases, and the input test cases into a dataset.

**Final Checks.** During experimental validation, we found and fixed multiple faulty problem description argument parsing functions. We then saved these corrected version as our final dataset.

**Additional Dataset Details.** In our experiments, we instruct LLMs to return their outputs. We maintain a wrapper script that executes the function $f(...)$ for each input, obtains the returned result, and then serializes the object as well as its type (recursively) into a string. We then utilize these strings as the basis for our semantic comparisons. For each generation, we execute all test cases and capture the resulting outputs.

Because LLMs often generate natural language to accompany generated programs, extracting programs from the generations is a non-trivial task, especially for pre-trained models. We developed an extraction utility that extracts not only the target function $f(...)$, but also any helper functions and imports that may be relevant. To safely execute programs at scale, we perform all execution inside an isolated DOCKER container to prevent adverse consequences of blindly executing LLM outputs.

### A.4 Raw Experimental Results for All Experiments

In Table 5 and Table 6, we include all our raw results used in our analysis.

### A.5 Additional Information on the Syntactic Diversity Metric

We extract and process all Abstract Syntax Trees (ASTs) using Python's AST library with Python version 3.12.0, and report metrics for subtrees of height 4. Because syntactically incorrect programs do not parse, we can only calculate this metric over the subset of syntactically correct generations.

### A.6 Prompts Used in Experiments

In Figure 7, Figure 8, and Figure 9, we show the prompting templates we use across all code experiments.

| Model | Prompt | Validity | Semantic | Lexical | Neural |
|---|---|---|---|---|---|
| Llama-3.1-70B-Instruct | Zero-Shot | 69.42 | 11.32 | 97.72 | 29.12 |
| Llama-3.1-70B-Instruct | Two-Shot | 36.28 | 6.02 | 98.56 | 49.98 |
| Llama-3.1-70B-Instruct | Two-Shot CoT | 33.15 | 5.24 | 98.83 | 52.69 |
| Llama-3.1-Tulu-3-70B | Zero-Shot | 91.95 | 15.95 | 98.92 | 21.20 |
| Llama-3.1-Tulu-3-70B | Two-Shot | 90.31 | 18.63 | 98.90 | 26.17 |
| Llama-3.1-Tulu-3-70B | Two-Shot CoT | 89.71 | 19.63 | 99.04 | 30.73 |
| Llama-3.1-Tulu-3-70B-DPO | Zero-Shot | 90.18 | 17.87 | 99.05 | 24.92 |
| Llama-3.1-Tulu-3-70B-DPO | Two-Shot | 90.10 | 17.65 | 99.02 | 24.86 |
| Llama-3.1-Tulu-3-70B-DPO | Two-Shot CoT | 89.39 | 20.11 | 98.95 | 32.27 |
| Llama-3.1-Tulu-3-70B-SFT | Zero-Shot | 68.33 | 22.83 | 98.42 | 56.75 |
| Llama-3.1-Tulu-3-70B-SFT | Two-Shot | 58.09 | 18.22 | 98.71 | 57.09 |
| Llama-3.1-Tulu-3-70B-SFT | Two-Shot CoT | 59.98 | 17.17 | 98.90 | 51.50 |
| Llama-3.1-70B | Zero-Shot | 56.12 | 17.97 | 96.51 | 59.86 |
| Llama-3.1-70B | Two-Shot | 40.72 | 10.07 | 98.13 | 67.81 |
| Llama-3.1-70B | Two-Shot CoT | 52.35 | 17.76 | 98.07 | 69.17 |
| Llama-3.1-8B-Instruct | Zero-Shot | 37.28 | 5.99 | 98.13 | 47.89 |
| Llama-3.1-8B-Instruct | Two-Shot | 24.11 | 2.81 | 99.24 | 51.81 |
| Llama-3.1-8B-Instruct | Two-Shot CoT | 24.31 | 3.05 | 99.14 | 53.22 |
| Llama-3.1-Tulu-3.1-8B | Zero-Shot | 90.55 | 17.83 | 98.89 | 24.17 |
| Llama-3.1-Tulu-3.1-8B | Two-Shot | 87.77 | 21.40 | 98.71 | 34.59 |
| Llama-3.1-Tulu-3.1-8B | Two-Shot CoT | 92.37 | 18.56 | 98.87 | 21.84 |
| Llama-3.1-Tulu-3-8B | Zero-Shot | 90.72 | 18.96 | 99.05 | 24.83 |
| Llama-3.1-Tulu-3-8B | Two-Shot | 85.47 | 19.08 | 98.87 | 33.14 |
| Llama-3.1-Tulu-3-8B | Two-Shot CoT | 91.11 | 18.49 | 98.97 | 26.63 |
| Llama-3.1-Tulu-3-8B-DPO | Zero-Shot | 88.84 | 18.13 | 98.84 | 30.67 |
| Llama-3.1-Tulu-3-8B-DPO | Two-Shot | 86.59 | 19.84 | 98.82 | 34.71 |
| Llama-3.1-Tulu-3-8B-DPO | Two-Shot CoT | 89.83 | 19.51 | 98.94 | 29.30 |
| Llama-3.1-Tulu-3-8B-SFT | Zero-Shot | 58.60 | 19.76 | 98.47 | 60.15 |
| Llama-3.1-Tulu-3-8B-SFT | Two-Shot | 55.41 | 18.75 | 98.70 | 58.21 |
| Llama-3.1-Tulu-3-8B-SFT | Two-Shot CoT | 57.70 | 19.55 | 98.81 | 57.43 |
| Llama-3.1-8B | Zero-Shot | 32.20 | 9.85 | 96.39 | 69.74 |
| Llama-3.1-8B | Two-Shot | 28.69 | 6.31 | 98.68 | 71.12 |
| Llama-3.1-8B | Two-Shot CoT | 37.89 | 11.80 | 98.57 | 74.33 |
| Llama-2-70b-chat-hf | Zero-Shot | 30.37 | 8.05 | 97.76 | 64.03 |
| Llama-2-70b-chat-hf | Two-Shot | 23.02 | 3.86 | 99.20 | 63.88 |
| Llama-2-70b-chat-hf | Two-Shot CoT | 24.64 | 4.34 | 99.24 | 65.06 |
| tulu-2-70b | Zero-Shot | 75.60 | 17.52 | 98.40 | 51.31 |
| tulu-2-70b | Two-Shot | 71.17 | 14.54 | 98.74 | 52.00 |
| tulu-2-70b | Two-Shot CoT | 73.37 | 17.88 | 98.69 | 54.56 |
| tulu-2-dpo-70b | Zero-Shot | 78.64 | 15.52 | 98.73 | 44.07 |
| tulu-2-dpo-70b | Two-Shot | 75.10 | 16.63 | 98.86 | 47.46 |
| tulu-2-dpo-70b | Two-Shot CoT | 78.60 | 17.68 | 98.84 | 49.00 |
| Llama-2-70b-hf | Zero-Shot | 22.87 | 7.11 | 96.85 | 75.34 |
| Llama-2-70b-hf | Two-Shot | 21.30 | 4.93 | 98.71 | 75.63 |
| Llama-2-70b-hf | Two-Shot CoT | 29.24 | 8.90 | 98.59 | 77.89 |
| Llama-2-7b-chat-hf | Zero-Shot | 22.39 | 4.98 | 98.34 | 67.20 |
| Llama-2-7b-chat-hf | Two-Shot | 16.54 | 2.44 | 99.48 | 67.81 |
| Llama-2-7b-chat-hf | Two-Shot CoT | 17.28 | 2.77 | 99.45 | 69.08 |
| tulu-2-7b | Zero-Shot | 53.99 | 12.95 | 98.07 | 66.84 |
| tulu-2-7b | Two-Shot | 49.64 | 11.01 | 98.56 | 67.74 |
| tulu-2-7b | Two-Shot CoT | 51.51 | 12.86 | 98.47 | 69.61 |
| tulu-2-dpo-7b | Zero-Shot | 64.67 | 13.41 | 98.49 | 58.75 |
| tulu-2-dpo-7b | Two-Shot | 59.52 | 12.74 | 98.74 | 60.59 |
| tulu-2-dpo-7b | Two-Shot CoT | 62.36 | 14.19 | 98.67 | 62.58 |
| Llama-2-7b-hf | Zero-Shot | 19.71 | 4.83 | 97.60 | 77.03 |
| Llama-2-7b-hf | Two-Shot | 17.70 | 3.44 | 99.08 | 77.63 |
| Llama-2-7b-hf | Two-Shot CoT | 23.52 | 5.98 | 98.93 | 79.62 |

Table 5: Raw Results for All Natural Language Experiments

| Model | Prompt | Validity | Semantic | Lexical | Syntactic | Neural |
|---|---|---|---|---|---|---|
| Llama-3.1-70B-Instruct | Zero-Shot | 37.36 | 32.96 | 67.19 | 71.98 | 5.60 |
| Llama-3.1-70B-Instruct | Two-Shot | 11.89 | 11.40 | 66.47 | 65.19 | 4.51 |
| Llama-3.1-70B-Instruct | Two-Shot CoT | 14.29 | 13.48 | 66.77 | 67.39 | 4.97 |
| Llama-3.1-Tulu-3-70B | Zero-Shot | 39.55 | 32.41 | 73.91 | 78.09 | 8.88 |
| Llama-3.1-Tulu-3-70B | Two-Shot | 41.49 | 33.22 | 72.76 | 75.33 | 4.43 |
| Llama-3.1-Tulu-3-70B | Two-Shot CoT | 40.88 | 32.85 | 73.26 | 76.69 | 8.31 |
| Llama-3.1-Tulu-3-70B-DPO | Zero-Shot | 38.57 | 32.93 | 75.37 | 79.81 | 9.54 |
| Llama-3.1-Tulu-3-70B-DPO | Two-Shot | 38.27 | 32.83 | 74.45 | 78.27 | 6.78 |
| Llama-3.1-Tulu-3-70B-DPO | Two-Shot CoT | 39.18 | 33.81 | 74.89 | 78.93 | 9.07 |
| Llama-3.1-Tulu-3-70B-SFT | Zero-Shot | 37.36 | 32.96 | 67.19 | 71.98 | 5.60 |
| Llama-3.1-Tulu-3-70B-SFT | Two-Shot | 35.48 | 29.37 | 68.76 | 71.38 | 5.23 |
| Llama-3.1-Tulu-3-70B-SFT | Two-Shot CoT | 36.42 | 31.12 | 68.48 | 72.15 | 5.40 |
| Llama-3.1-70B | Zero-Shot | 20.18 | 24.02 | 54.05 | 55.47 | 3.51 |
| Llama-3.1-70B | Two-Shot | 13.28 | 14.70 | 57.87 | 57.00 | 3.46 |
| Llama-3.1-70B | Two-Shot CoT | 15.17 | 18.00 | 57.25 | 58.00 | 3.64 |
| Llama-3.1-8B-Instruct | Zero-Shot | 15.47 | 18.98 | 56.40 | 60.12 | 3.41 |
| Llama-3.1-8B-Instruct | Two-Shot | 8.54 | 10.37 | 56.09 | 57.54 | 3.22 |
| Llama-3.1-8B-Instruct | Two-Shot CoT | 9.94 | 12.70 | 55.63 | 58.68 | 3.35 |
| Llama-3.1-Tulu-3.1-8B | Zero-Shot | 37.67 | 32.41 | 73.32 | 77.66 | 8.91 |
| Llama-3.1-Tulu-3.1-8B | Two-Shot | 37.06 | 32.67 | 72.45 | 76.10 | 6.83 |
| Llama-3.1-Tulu-3.1-8B | Two-Shot CoT | 38.57 | 33.70 | 73.19 | 77.53 | 8.73 |
| Llama-3.1-Tulu-3-8B | Zero-Shot | 38.27 | 32.93 | 74.52 | 78.64 | 9.22 |
| Llama-3.1-Tulu-3-8B | Two-Shot | 36.73 | 31.85 | 73.19 | 76.69 | 6.20 |
| Llama-3.1-Tulu-3-8B | Two-Shot CoT | 37.97 | 32.85 | 73.91 | 77.83 | 8.66 |
| Llama-3.1-Tulu-3-8B-DPO | Zero-Shot | 37.06 | 32.41 | 74.06 | 78.28 | 8.91 |
| Llama-3.1-Tulu-3-8B-DPO | Two-Shot | 36.42 | 31.70 | 73.19 | 76.69 | 6.83 |
| Llama-3.1-Tulu-3-8B-DPO | Two-Shot CoT | 37.67 | 32.56 | 73.75 | 77.53 | 8.44 |
| Llama-3.1-Tulu-3-8B-SFT | Zero-Shot | 35.18 | 29.37 | 68.17 | 71.38 | 5.23 |
| Llama-3.1-Tulu-3-8B-SFT | Two-Shot | 34.85 | 29.67 | 68.01 | 71.18 | 5.40 |
| Llama-3.1-Tulu-3-8B-SFT | Two-Shot CoT | 35.48 | 30.11 | 68.32 | 71.78 | 5.57 |
| Llama-3.1-8B | Zero-Shot | 11.89 | 15.99 | 47.23 | 48.41 | 2.78 |
| Llama-3.1-8B | Two-Shot | 9.64 | 12.41 | 51.05 | 51.11 | 2.73 |
| Llama-3.1-8B | Two-Shot CoT | 11.28 | 14.70 | 50.43 | 51.51 | 2.91 |
| Llama-2-70b-chat-hf | Zero-Shot | 11.28 | 15.84 | 52.30 | 55.87 | 3.30 |
| Llama-2-70b-chat-hf | Two-Shot | 6.43 | 8.52 | 51.82 | 53.57 | 3.11 |
| Llama-2-70b-chat-hf | Two-Shot CoT | 7.82 | 10.67 | 52.14 | 54.36 | 3.25 |
| tulu-2-70b | Zero-Shot | 31.86 | 29.52 | 64.74 | 68.35 | 5.06 |
| tulu-2-70b | Two-Shot | 29.27 | 26.22 | 64.58 | 66.62 | 4.57 |
| tulu-2-70b | Two-Shot CoT | 30.57 | 28.52 | 64.90 | 67.79 | 4.95 |
| tulu-2-dpo-70b | Zero-Shot | 33.26 | 30.41 | 67.03 | 70.28 | 5.40 |
| tulu-2-dpo-70b | Two-Shot | 31.25 | 27.70 | 66.71 | 68.35 | 4.79 |
| tulu-2-dpo-70b | Two-Shot CoT | 32.56 | 29.37 | 66.87 | 69.52 | 5.23 |
| Llama-2-70b-hf | Zero-Shot | 8.24 | 12.11 | 44.15 | 45.33 | 2.59 |
| Llama-2-70b-hf | Two-Shot | 6.43 | 9.96 | 46.58 | 47.02 | 2.54 |
| Llama-2-70b-hf | Two-Shot CoT | 8.24 | 11.26 | 46.27 | 47.42 | 2.68 |
| Llama-2-7b-chat-hf | Zero-Shot | 8.85 | 12.56 | 49.21 | 52.31 | 3.03 |
| Llama-2-7b-chat-hf | Two-Shot | 5.82 | 7.78 | 48.58 | 50.18 | 2.84 |
| Llama-2-7b-chat-hf | Two-Shot CoT | 7.22 | 9.67 | 48.89 | 51.11 | 2.97 |
| tulu-2-7b | Zero-Shot | 24.39 | 24.46 | 58.25 | 61.63 | 4.13 |
| tulu-2-7b | Two-Shot | 22.56 | 21.70 | 58.09 | 59.89 | 3.68 |
| tulu-2-7b | Two-Shot CoT | 23.78 | 23.41 | 58.41 | 60.89 | 3.98 |
| tulu-2-dpo-7b | Zero-Shot | 26.83 | 26.07 | 61.08 | 64.36 | 4.46 |
| tulu-2-dpo-7b | Two-Shot | 24.39 | 23.41 | 60.61 | 62.43 | 3.98 |
| tulu-2-dpo-7b | Two-Shot CoT | 25.91 | 25.19 | 60.92 | 63.56 | 4.30 |
| Llama-2-7b-hf | Zero-Shot | 6.73 | 10.22 | 41.72 | 42.70 | 2.38 |
| Llama-2-7b-hf | Two-Shot | 5.52 | 8.37 | 43.84 | 44.33 | 2.32 |
| Llama-2-7b-hf | Two-Shot CoT | 6.73 | 9.52 | 43.68 | 44.53 | 2.46 |

Table 6: Raw Results for General Code Experiments

## A.7 Constrained Generation

We implement a validity oracle that only accepts lists of integers with a maximum length of 1000. The constraints are illustrated in the following code snippet, where *output* denotes the LLM generation:

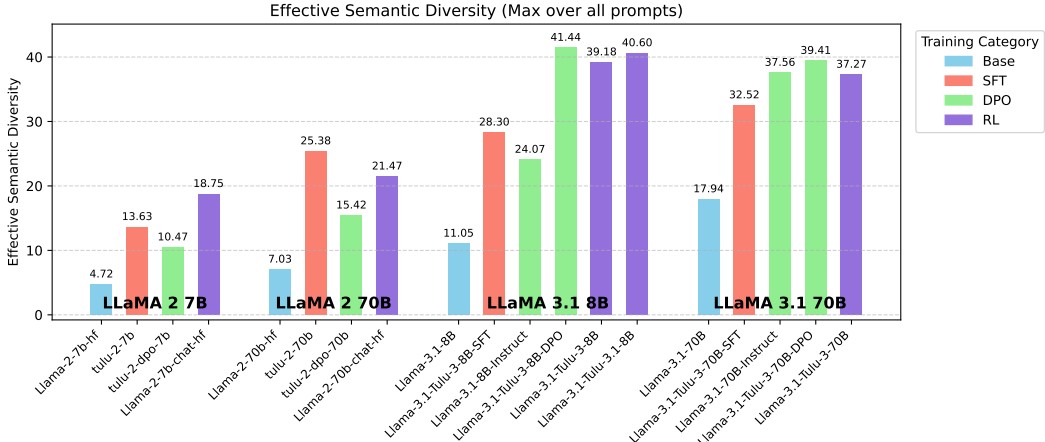

Figure 5: **Effective semantic diveresity scores for all 19 models evaluated in our experiments, grouped by model family.** Each bar is color-coded according to the post-training method, as categorized in Table 1

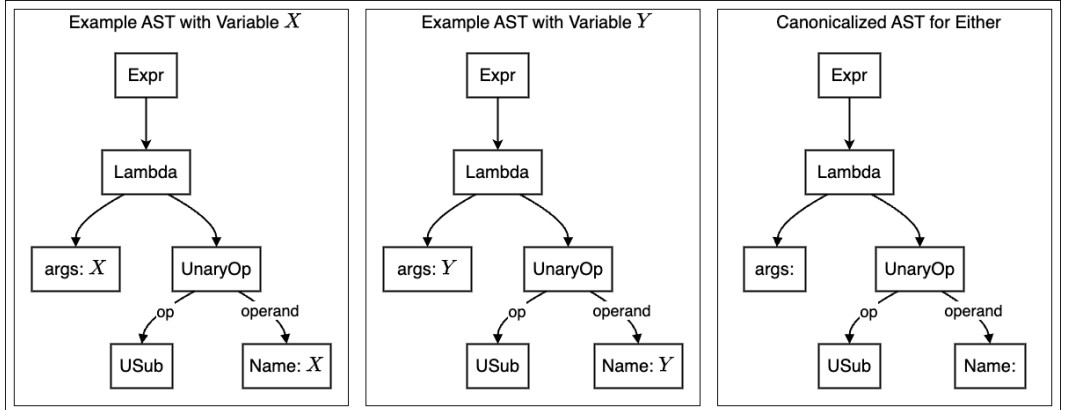

Figure 6: An Example of Canonicalizing an Abstract Syntax subtree used in the Distinct-CAST metric. The expression under consideration is for a simple lambda expression that negates a given variable. The first two ASTs are not equal because of the usage of the variables $X$ and $Y$, respectively, even though they are alpha-equivalent expressions. The AST on the far right canonicalizes identifier names such as arguments and variables so that both expressions would be equivalent.

```
assert isinstance(output, list)
assert all(isinstance(x, int) for x in output)
assert len(output) < 1000
```

Furthermore, we modify all prompts to instruct LLM to comply with these constraints. We show results in Table 7. In general, our findings are consistent with our previous results. The most notable difference is an increase in both statistical significance and effect size for semantic diversity when comparing DPO-tuned models to RL-tuned models for code generation. While this difference is of interest, it does not affect our main conclusions.

## A.8 Natural Language Experiments

In this section, we extend our experiments to natural language tasks, including creative writing, argumentative writing, and brainstorming. Specifically, we obtained 10 prompts each for argumentative writing and creative writing from the CoAuthor dataset (Lee et al.,

```
{problem_description}

Now please implement the function f; do not return anything, the function f
↪  should print the result of the operation.
It should terminate within 30 seconds.
```

Figure 7: **Zero-Shot Prompt**: Our template for our zero-shot prompt, where the problem description would be input inside the curly braces.

| Comparison | Validity (Quality) | | | Semantic Diversity | | |
|---|---|---|---|---|---|---|
| | W (p) | ES (d) | Winner | W (p) | ES (d) | Winner |
| Base vs. Inst. | < 0.001 | 1.23 | Inst. | < 0.001 | 1.27 | Inst. |
| Small vs. Lg. | 0.001 | 0.35 | Lg. | 0.001 | 0.35 | Lg. |
| SFT vs. Pref. | < 0.001 | 1.12 | Pref. | < 0.001 | 1.04 | Pref. |
| SFT vs. DPO | 0.064 | 0.69 | DPO | 0.064 | 0.64 | DPO |
| SFT vs. RL | 0.004 | 3.28 | RL | 0.004 | 3.14 | RL |
| DPO vs. RL | 0.237 | -0.19 | DPO | 0.039 | -0.44 | DPO |

| Comparison | Syntactic Diversity | | | Lexical Diversity | | | Raw Neural Diversity | | |
|---|---|---|---|---|---|---|---|---|---|
| | W (p) | ES (d) | Winner | W (p) | ES (d) | Winner | W (p) | ES (d) | Winner |
| Base vs. Inst. | < 0.001 | -0.88 | Base | 0.701 | -0.16 | Base | 0.010 | -1.61 | Base |
| Small vs. Lg. | 0.248 | 0.14 | Lg. | 0.645 | 0.02 | Lg. | 0.036 | 0.16 | Lg. |
| SFT vs. Pref. | < 0.001 | -0.95 | SFT | < 0.001 | -0.60 | SFT | < 0.001 | -2.24 | SFT |
| SFT vs. DPO | < 0.001 | -0.64 | SFT | 0.007 | -0.46 | SFT | < 0.001 | -1.77 | SFT |
| SFT vs. RL | 0.004 | -2.18 | SFT | 0.008 | -1.04 | SFT | 0.004 | -2.98 | SFT |
| DPO vs. RL | 0.426 | -0.15 | DPO | 0.301 | -0.37 | DPO | 1.000 | 0.05 | RL |

Table 7: **Constrained Generation Model Comparison Results.**

2022), which has been previously used in the literature (Padmakumar & He, 2024). We also manually curated a dataset of 10 brainstorming prompts designed to reasonably reflect creative assistance tasks for an LLM-based assistant.

We employed GPT-4.1-mini as a judge for both the quality and diversity of generations, selecting it to balance robustness with our API budget. Separate evaluation prompts were constructed for each of the three tasks to specify the criteria for quality and diversity assessment. For example, the criteria for creative writing tasks were as follows:

**Creative Quality Criteria**    Consider the following rubric when evaluating:

1. Overall, holistic, and cohesive readability of the story (not merely a compilation of elements).
2. Relevance of the story to the provided prompt.
3. Use of key narrative elements—vocabulary choice, imagery, setting, themes, dialogue, characterisation, and point of view.
4. Structural elements and presentation demonstrating control over spelling, grammar, punctuation, paragraphing, and formatting.
5. Overall plot logic, including hook, conflict, initial crisis, rising and falling action, and denouement/resolution.
6. Creativity, innovation, and originality—credibility, introduction of new knowledge, and avoidance of clichés and derivative tropes.

**Creative Diversity Criteria**    Consider the following criteria when evaluating similarity:

```
### Input Description:
1. An integer \( N \) (1  \( N \)  10000), representing some quantity or size.
### Example Input:
```

1000
```

### Function Signature:
Write a function `f(N)` that takes in the input.
```python
def f(N: int):
    '''

    N: an integer
    '''
Now please implement the function f; do not return anything, the function f
↪   should print the result of the operation.
It should terminate within 30 seconds.
def f(N: int):
    print(n**2)
### Input Description:
1. A floating point number \( N \) (1  \( N \)  10000), representing some
↪   quantity or size.
### Example Input:
```

143.23
```

### Function Signature:
Write a function `f(N)` that takes in the input.
```python
def f(N: float):
    '''

    N: a float
    '''
Now please implement the function f; do not return anything, the function f
↪   should print the result of the operation.
It should terminate within 30 seconds.
def f(N: float):
    i = 0
    while N > 1:
        N = N / 2
        i += 1
    print(i)
{problem_description}
Now please implement the function f; do not return anything, the function f
↪   should print the result of the operation.
It should terminate within 30 seconds.
```

Figure 8: **Two-Shot Prompt**: Our template for our two-shot prompt, where the problem description would be input near the end inside the curly braces.

1. **Semantic Overlap:** Do the responses share similar underlying themes, ideas, narrative elements, or emotional content?

2. **Thematic Consistency:** Do both responses explore similar themes or motifs?

Our prompts instructed the model to assign a score from 1–10 for each element. Thus, the first prompt had a maximum possible score of 60, and the second had a maximum possible score of 20. We used OpenAI's grammar-constrained decoding to ensure integer outputs, combined with a chain-of-thought reasoning component to improve robustness. All scores were normalized by the maximum possible score.

For diversity evaluation, we sub-sampled 32 pairs from all possible pairs with replacement and asked the LLM judge to score the similarity between the two generations. The diversity score was computed as:

$$\text{Diversity Score} = 1 - \text{Sim}(g_i^j, g_i^k).$$

**Effective Semantic Diversity**    Because we could only calculate a pairwise diversity metric for natural language, we applied Equation (3) from the paper to compute effective semantic diversity across sub-sampled pairs. For completeness, we used two methods to measure pairwise effective semantic diversity that we considered reasonable and consistent with our intended intuition.

The pairwise diversity metric from Equation (3) is defined as:

$$ESD_{\text{pair}}(P_i) = \frac{1}{\binom{K}{2}} \sum_{j<k} d_{\text{sem}}(g_i^j, g_i^k),$$

where $d_{\text{sem}} : \mathcal{G} \times \mathcal{G} \rightarrow \{0,1\}$ is given by:

$$d_{\text{sem}}(g_i^j, g_i^k) = \begin{cases} 0 & \text{if either generation is invalid,} \\ 0 & \text{if both are valid and semantically identical,} \\ 1 & \text{if both are valid and semantically distinct.} \end{cases}$$

For natural language, we adapt this definition using **Hard Thresholding**:

$$d_{\text{sem}}(g_i^j, g_i^k) = \begin{cases} 1 & \text{if } LLM_{\text{div}}(g_i^j, g_i^k) > 0.5 \text{ and } LLM_{\text{qual}}(g_i^j) > 0.5, \\ 0 & \text{otherwise.} \end{cases}$$

We also adapt it using **Soft Weighting**:

$$d_{\text{sem}}(g_i^j, g_i^k) = LLM_{\text{div}}(g_i^j, g_i^k) \times LLM_{\text{qual}}(g_i^j) \times LLM_{\text{qual}}(g_i^k).$$

We present results for the creative writing task in Table 8, the argumentative writing task in Table 9, and the brainstorming task in Table 10. Overall, our findings for natural language tasks largely mirror those observed for code. Across all experiments, we consistently observe that post-training is associated with higher effective semantic diversity relative to base models. Furthermore, RL-tuned models generally exhibit higher effective semantic diversity than SFT-tuned models, often more markedly than DPO-tuned models. In argumentative and creative writing, larger models tend to achieve higher effective semantic diversity than smaller models, although this trend is less consistent in the creative brainstorming task. Importantly, in these natural language settings, we find little to no evidence that post-training strategies induce lexical mode-collapse. Moreover, when quality is not taken into account, Raw Neural Diversity (as assessed by the LLM-judge) tends to be higher for less aggressive post-training regimes (e.g., base models outperform instruction-tuned models, and SFT models outperform RL-tuned models). However, once quality is incorporated into the evaluation, this effect can be fully reversed, consistent with our findings for code. Consequently, more aggressive post-training regimes, such as PPO, are ultimately associated with higher effective semantic diversity.

These experiments also underscore potential advantages of using code to evaluate effective semantic diversity. For code, our validity criteria (quality threshold) required both syntactic correctness and successful execution of all test cases without runtime errors; in contrast, for natural language, LLM-judge scores are less interpretable and may be susceptible to reward hacking. Additionally, code execution for diversity and quality assessment completes relatively quickly and incurs minimal cost, whereas LLM-judge evaluation can be computationally and financially expensive, necessitating careful selection of sample sizes

to control API usage. Finally, in an environment where increasingly powerful models are being developed, using code to evaluate the diversity–quality trade-off may offer distinct advantages over relying on weaker models to evaluate potentially stronger models.

| Comparison | Validity (Quality) | | | Soft Effective Semantic Diversity | | | Hard Effective Semantic Diversity | | |
|---|---|---|---|---|---|---|---|---|---|
| | W (p) | ES (d) | Winner | W (p) | ES (d) | Winner | W (p) | ES (d) | Winner |
| Base vs. Inst. | < 0.001 | 1.69 | Inst. | < 0.001 | 0.80 | Inst. | < 0.001 | 1.67 | Inst. |
| Small vs. Lg. | < 0.001 | 0.37 | Lg. | 0.004 | 0.47 | Lg. | 0.020 | 0.41 | Lg. |
| SFT vs. Pref. | < 0.001 | 2.13 | Pref. | 0.010 | 0.62 | Pref. | 0.050 | 0.58 | Pref. |
| SFT vs. DPO | < 0.001 | 1.33 | DPO | 0.092 | 0.42 | DPO | 0.176 | 0.39 | DPO |
| SFT vs. RL | 0.004 | 5.80 | RL | 0.074 | 0.99 | RL | 0.203 | 0.83 | RL |
| DPO vs. RL | 0.008 | 0.40 | RL | 0.129 | -0.45 | DPO | 0.203 | -0.29 | DPO |

| Comparison | Lexical Diversity | | | Raw Neural Diversity | | |
|---|---|---|---|---|---|---|
| | W (p) | ES (d) | Winner | W (p) | ES (d) | Winner |
| Base vs. Inst. | < 0.001 | 1.30 | Inst. | < 0.001 | -3.21 | Base |
| Small vs. Lg. | 0.202 | -0.13 | Lg. | 0.010 | -0.20 | Small |
| SFT vs. Pref. | < 0.001 | 0.95 | Pref. | < 0.001 | -2.38 | SFT |
| SFT vs. DPO | < 0.001 | 0.90 | DPO | < 0.001 | -1.51 | SFT |
| SFT vs. RL | 0.004 | 1.70 | RL | 0.004 | -5.57 | SFT |
| DPO vs. RL | 0.570 | -0.15 | DPO | 0.098 | -0.37 | DPO |

Table 8: **Creative Writing Model Comparison Results.**

| Comparison | Validity (Quality) | | | Soft Effective Semantic Diversity | | | Hard Effective Semantic Diversity | | |
|---|---|---|---|---|---|---|---|---|---|
| | W (p) | ES (d) | Winner | W (p) | ES (d) | Winner | W (p) | ES (d) | Winner |
| Base vs. Inst. | < 0.001 | 1.93 | Inst. | < 0.001 | 0.86 | Inst. | < 0.001 | 1.69 | Inst. |
| Small vs. Lg. | 0.248 | 0.12 | Lg. | 0.026 | 0.54 | Lg. | 0.044 | 0.37 | Lg. |
| SFT vs. Pref. | < 0.001 | 1.55 | Pref. | 0.033 | 0.49 | Pref. | 0.032 | 0.41 | Pref. |
| SFT vs. DPO | < 0.001 | 1.07 | DPO | 0.213 | 0.26 | DPO | 0.266 | 0.22 | DPO |
| SFT vs. RL | 0.008 | 2.40 | RL | 0.129 | 1.07 | RL | 0.129 | 1.14 | RL |
| DPO vs. RL | 0.020 | 0.22 | RL | 0.263 | 0.10 | RL | 0.301 | 0.12 | RL |

| Comparison | Lexical Diversity | | | Raw Neural Diversity | | |
|---|---|---|---|---|---|---|
| | W (p) | ES (d) | Winner | W (p) | ES (d) | Winner |
| Base vs. Inst. | 0.033 | 0.16 | Inst. | < 0.001 | -3.07 | Base |
| Small vs. Lg. | 0.010 | -0.33 | Small | 0.594 | -0.00 | Small |
| SFT vs. Pref. | 0.919 | -0.21 | SFT | < 0.001 | -1.90 | SFT |
| SFT vs. DPO | 0.970 | -0.25 | SFT | < 0.001 | -1.43 | SFT |
| SFT vs. RL | 0.910 | -0.14 | SFT | 0.004 | -3.11 | SFT |
| DPO vs. RL | 0.426 | 0.21 | RL | 0.359 | -0.20 | DPO |

Table 9: **Argumentative Writing Model Comparison Results.**

| Comparison | Validity (Quality) | | | Soft Effective Semantic Diversity | | | Hard Effective Semantic Diversity | | |
|---|---|---|---|---|---|---|---|---|---|
| | W (p) | ES (d) | Winner | W (p) | ES (d) | Winner | W (p) | ES (d) | Winner |
| Base vs. Inst. | < 0.001 | 1.27 | Inst. | < 0.001 | 1.05 | Inst. | 0.002 | 0.86 | Inst. |
| Small vs. Lg. | 0.859 | -0.32 | Small | 0.645 | -0.34 | Small | 0.546 | -0.23 | Small |
| SFT vs. Pref. | 0.013 | 0.37 | Pref. | 0.137 | 0.15 | Pref. | 0.919 | -0.02 | SFT |
| SFT vs. DPO | 0.339 | 0.14 | DPO | 0.970 | 0.01 | DPO | 0.151 | -0.10 | SFT |
| SFT vs. RL | 0.020 | 0.72 | RL | 0.039 | 0.34 | RL | 0.426 | 0.09 | RL |
| DPO vs. RL | 0.055 | 0.24 | RL | 0.164 | 0.13 | RL | 0.203 | 0.12 | RL |

| Comparison | Lexical Diversity | | | Raw Neural Diversity | | |
|---|---|---|---|---|---|---|
| | W (p) | ES (d) | Winner | W (p) | ES (d) | Winner |
| Base vs. Inst. | < 0.001 | 1.20 | Inst. | < 0.001 | -3.16 | Base |
| Small vs. Lg. | 0.036 | -0.58 | Small | < 0.001 | -0.48 | Small |
| SFT vs. Pref. | 0.759 | -0.13 | SFT | < 0.001 | -1.64 | SFT |
| SFT vs. DPO | 0.176 | -0.31 | SFT | < 0.001 | -1.32 | SFT |
| SFT vs. RL | 0.301 | 0.16 | RL | 0.008 | -2.19 | SFT |
| DPO vs. RL | 0.027 | 0.45 | RL | 0.734 | 0.36 | RL |

Table 10: **Brainstorming / Creative Assistance Model Comparison Results.**

```
### Input Description:
1. An integer \( N \) (1  \( N \)   10000), representing some quantity or size.
### Example Input:
```
1000
```

### Function Signature:
Write a function `f(N)` that takes in the input.
```python
def f(N: int):
    '''

    N: an integer
    '''
Now please implement the function f; do not return anything, the function f
↪   should print the result of the operation.
It should terminate within 30 seconds. First describe the function you would
↪   write, then implement it.
The following function will print out the square of the input number. We will
↪   take the square using the ** operator in Python within the print
↪   statement.
def f(N: int):
    print(n**2)
### Input Description:
1. A floating point number \( N \) (1  \( N \)   10000), representing some
↪   quantity or size.
### Example Input:
```
143.23
```

### Function Signature:
Write a function `f(N)` that takes in the input.
```python
def f(N: float):
    '''

    N: a float
    '''
Now please implement the function f; do not return anything, the function f
↪   should print the result of the operation.
It should terminate within 30 seconds. First describe the function you would
↪   write, then implement it.
The following function will calculate the number of times the input number can
↪   be divided by 2 before it becomes less than 1. We will increment a counter
↪   variable i each time we divide the number by 2 inside a while loop.
def f(N: float):
    i = 0
    while N > 1:
        N = N / 2
        i += 1
    print(i)
{problem_description}
Now please implement the function f; do not return anything, the function f
↪   should print the result of the operation.
It should terminate within 30 seconds. First describe the function you would
↪   write, then implement it.
```

Figure 9: **Two-Shot Chain-of-Thought Prompt**: Our template for our two-shot Chain-of-Thought prompt, where the problem description would be input near the end inside the curly braces.

