# OpenReview forum: "Evaluating the Diversity and Quality of LLM Generated Content"
_colmweb.org/COLM/2025/Conference — COLM 2025_

### Official Review · Reviewer_rZeC · 2025-04-28

**Rating:** 6
**Confidence:** 4
**Ethics Flag:** 1

**Summary:**

This paper introduces a framework for evaluating the effective semantic diversity of outputs generated by large language models (LLMs), focusing on diversity among responses that meet certain quality thresholds. Specifically, they use the program generation task as the open-ended text generation task to evaluate, in order to avoid using LLMs to evaluate LLMs for diversity. The study reveals that preference-tuning techniques, such as RLHF methods like PPO and GRPO, while reducing lexical and syntactic diversity, surprisingly enhance effective semantic diversity compared to supervised fine-tuning (SFT) or base models. This enhancement arises not from increasing diversity within high-quality outputs but from generating a higher proportion of high-quality outputs overall. The research also uncovers that smaller models are more parameter-efficient in producing unique content within a fixed sampling budget.

**Reasons To Accept:**

* The diversity of LLM generated content is an interesting topic.

* The idea of using the program generation task as the open-ended text generation task to evaluate diversity is an brilliant. Since we have many tools like program execution and abstract syntax tree to subjectively evaluate the diversity and effectiveness of generated content on this task.

* They evaluate various LLMs with different settings (base/SFT/DPO/etc.), and report the statistical value like p-values and Cohen’s D to show the significance of some assumptions about the diversity of LLMs' generation.

* The findings, like "RLHF methods, while reducing lexical and syntactic diversity, enhance effective semantic diversity compared to SFT", are potentially helpful for the researchers in the area of LLMs.

**Reasons To Reject:**

* Using the program generation task is an brilliant idea. However, it still brings the limitation that models' performances on program generation are not necessarily consistent to the general performances.

* The sec. 5, "Erimental Results" is a little bit short, lack more detailed explainations and discussions about the experimental results. Instead of the p-values and Cohen’s D, some tables/figures to show the absolute measurement of semantic diversity is also meaningful. (Figure-3 is not very clear, since it contains so many information.)

Minor problem:
* Figure-1 is a little bit ambiguous. It takes me a few minutes to figure out that the three model names (Llama 3.1 70B Instruct, etc.) are not aligned with the three sub-figures respectively.

---

> ### Author Response · Authors · 2025-06-01
>
> Thank you for your review! We appreciate your thoughts on our use of open-ended programs… that was a very subtle point in the paper but as researchers we are glad that using programs helped us avoid issues such as having to rely on LLMs to evaluate LLMs / using LLM-as-a-judge as an oracle for diversity: as these may be problematic.
>
> **Re: experimental results section**
>
> Thanks for your feedback! For COLM unfortunately we are unable to update the PDF during the rebuttal process, but we can do some of the following:
>
> 1. We will commit to adding these extra tables to the appendix and we can provide these detailed tables on OpenReview if you like.
> 2. After taking a look at the Experimental Results section, we also feel that the writing of this section is a little crammed-in. If we were accepted, we would commit to expanding this section, especially because we would have an extra page of content allowed. We would like to provide more assurances to you, but we're not sure what else we can do: if you like we could post a longer draft of this section on OpenReview before the rebuttal deadline.
>
> **Re: limitations of general insights**
>
> You bring up a tricky point and it seems like all reviewers and us as well, that it would be awesome if we could obtain more general results. As we explained to the other reviewers, for code we can use execution to understand semantics, but for natural language this is non-trivial. Automatic methods like LLM-as-a-judge may not be as robust as code, and we worried that if we used an LLM-as-a-judge that our paper would lack rigor and so we spent more effort on deeper analysis with programs.
>
> There seems to be some different viewpoints on this issue: reviewer p6Pc has pointed out that we should either change our language to be more focused on code and thinks the paper would stand on its own, or we could do a tiny natural language experiment to show how this could be extended to natural language. We'd like to do both: do you agree with p6Pc that if we did either of these, our paper is more likely to meet the threshold necessary for COLM?

---

> > ### Comment · Reviewer_rZeC · 2025-06-07
> >
> > From my perspective, either focusing on code generation (as commented by Reviewer 7jf8, this means "a major revision of this paper and fundamentally alter the framework of this paper"), or collecting more evidence and providing more general insights (to better support the title "Evaluating the Diversity and Quality of LLM Generated Content") can make the paper better.

---

> > > ### Author Response · Authors · 2025-06-10
> > >
> > > Thank you for your thoughtful review and for recognizing the value of using program generation to evaluate the quality and diversity of LLM generations.
> > >
> > > ### On Generalizability Beyond Code Generation
> > >
> > > Please see our response to all reviewers above, where we detail natural language experiments across 3 domains that generally align with our key findings in the manuscript. These experiments also highlight the advantages of using code execution as an oracle: it provides more objective validity and diversity checks, it avoids potential circularity of using LLMs to evaluate LLMs, and it may be computationally less expensive than LLM-judge approaches. For the final draft, we plan to re-run at least one of the natural language experiments with an order of magnitude more prompts and include these natural language experiments in the revised paper.
> > >
> > > ### On the Experimental Results Section
> > >
> > > We agree Section 5 needs expansion.
> > >
> > > For the final draft, we will expand this section with more detailed discussion. We hypothetically have an extra page available which would allow for this expansion.
> > >
> > > ### On Figure 1
> > >
> > > For a revised draft we will think critically on how we can improve the intuition and readability of Figure 1. Thank you again for the constructive feedback.

---

> > > > ### Author Response · Authors · 2025-06-11
> > > >
> > > > As the window closes, we’d like to ask if you have any remaining concerns. We forgot to mention that in the appendix in Figure 5 on Page 16 we have a bar chart that provides a more intuitive and visual representation of effective semantic diversity over all models. We hope this addresses some of your concerns regarding readability and we will continue to reflect on how we can make the results more intuitive to read using the extra page that would hypothetically be allowed.
> > > >
> > > > We believe our additional natural language experiments across creative writing, argumentative writing, and brainstorming tasks address your concern about generalizability beyond code generation, and we hope to implement the changes you have suggested in a revised draft (COLM does not allow updating the PDF during the rebuttal process).
> > > >
> > > > We understand and respect your perspective, and if we have addressed your concerns we would be grateful if you would consider revisiting your score. Thank you again for your time and your constructive review. We have appreciated this opportunity to improve our work based on your feedback.

---

### Official Review · Reviewer_7jf8 · 2025-05-09

**Rating:** 6
**Confidence:** 4
**Ethics Flag:** 1

**Summary:**

This paper focused on the influence of different preference-tuning methods on LMs' generation diversity. Through a new metric proposed, this paper obtained a viewpoint against consensus, that is, preference-tuning can bring beneficial influence on LMs' generative diversity. Several models and preference-tuning methods are compared in an open-ended programming task.

**Questions To Authors:**

- What further insights can this metric bring to the design of reinforcement learning algorithms for LMs?

**Reasons To Accept:**

This paper brought two valuable insights on the evaluation of generation diversity:
- Firstly, generation validity should be one of the necessary prerequisite conditions.
- Secondly, semantic diversity might be more valuable than syntactic and lexical diversity.

**Reasons To Reject:**

- Weak and Insufficient Empirical Setting
  - Although ideation and creative assistance are core targets of generation diversity (lines 23-25), this paper only evaluated their metric on a programming task, in which creativity and diversity are not as important as other tasks (i.e. creative writing, scientific hypothesis proposal and so on).
  - Considering the main contribution of this paper is the proposed metric, the sufficient empirical study on diverse tasks is particularly essential. Thus, I believe more tasks should be considered in the experiments.
- (Perhaps) Overclaim on the effectiveness to eliminate correlation with temperature
  - It is clear that the metric proposed does not demonstrate a “linear correlation" agaist temperature.
  - However, correlation represented by higher-order functions still exists in the proposed metric, that is, the score in the new metric is significantly higher in the middle (independent with the models and methods).

I look forward further discussion with the authors and will reconsider my assessment after my concerns are addressed.

---

> ### Author Response · Authors · 2025-06-01
>
> Thanks for your review! We'd like to address some of your highlighted concerns and make some clarifications.
>
> **On the potential overclaim on "eliminating the correlation with temperature"**
>
> I think there may be some confusion on our intent: the non-linear relationship between ESD and temperature is desired and validates the metric. If we have any wording that confuses this we'd want to fix that. In fact, this non-linear relationship you describe is actually intended.
>
> We include this figure to be like a sanity check: we expect there to be an "optimal point" with temperature sampling roughly between 0.6 - 1.2: too low and the LLM is too-deterministic and lacks diversity; too-high the LLM is random and incoherent. We want ESD to penalize excessively low and high temperature values and we find that it does! Whereas traditional diversity metrics that do not incorporate quality fail to reflect this. We think it would actually be problematic if our ESD metric did not have this inverse-quadratic relationship. So it sounds to us like you agree that the relationship holds, I think maybe there is just some confusion that we intend this as a feature and not a bug. If this is unclear in the manuscript, we would want to change this language and clarify this. If you feel we've addressed this as an issue, does that affect your assessment of the paper? Of course we understand you have some additional concerns as well.
>
> **On Only Evaluating on a Programming Task**
>
> We completely agree that it would be important to evaluate more domains, we mention this in lines 329-331 in the conclusion. We agree the language in the paper should be changed to be more specific to code. However, we sense there is some disagreement among reviewers: reviewer p6Pc claims that if we change the wording of the paper, they think the paper would be enough to stand on its own. If we are unable to complete some small NL experiments, we would commit to changing the language of the paper in its final draft.
>
> We want to emphasize that open-ended code generation is actually relevant: we reference the Llama3 paper [1] in lines 199-200 which created 2.7 million synthetic programs for post-training the Llama 3 herd of models. LLMs for software engineering is a very large domain, and because it is expensive to manually create high quality programing examples, creating diverse synthetic examples for programming is already an actively used technique. That being said, we agree that being able to additionally evaluate creative writing would be interesting and valuable.
>
> To explain our choice for using code-only: we worried that if we used an LLM judge for diversity and quality that it would not be rigorous enough for publication without more evidence showing this approach is appropriate, and manual human evaluation cannot scale to evaluating tens-of-thousands of generations. While we wouldn't use as enthusiastic wording, reviewer rZeC explains that "program generation task as the open-ended text generation task" is "brilliant" because we can use "tools like program execution and abstract syntax tree" to analyze diversity. We will try to follow the suggestion by reviewer p6Pc to do a tiny experiment for natural language using an LLM judge with the caveat that an LLM judge is not as rigorous as code execution and analysis.
>
> **Re: What further insights can this metric bring to the design of reinforcement learning algorithms for LMs?**
>
> We think that our dataset + metric can be used by research groups building RL-tuned LLMs to ensure that their post-training does not experience some sort of distribution collapse. As we point out in section 2, in theory over-optimizing for a reward model will lead to distribution collapse; however, KL-regularized RL should preserve diversity. We anticipate our dataset could be used with other benchmarks to help decide which RL algorithms are optimal and what KL-regularization penalty is optimal for *open-ended generation*.
>
> We also think that our dataset + metric will make it easier for other researchers to explore novel post-training algorithms that optimize for diversity. A recent example is DivPO [2] which proposes a DPO-inspired algorithm for optimizing diversity. In the paper, the researchers use toy-tasks for evaluation which are far less semantically sophisticated than using full programs, and their results may be strengthened by leveraging our dataset.
>
> In terms of other practical insights, the finding that DPO may have more lexical and syntactic diversity than PPO without a significant difference in semantic diversity may inspire different model use. For programming tasks where a consistent style may be desirable, PPO may be preferable. Whereas in creative writing where diverse style may be preferable, assuming the same results held, DPO-tuned models may be preferable.
>
> [1] Grattafiori, Aaron, et al. "The llama 3 herd of models." (2024).
>
> [2] Lanchantin, Jack, et al. "Diverse Preference Optimization."(2025).

---

> > ### Comment · Reviewer_7jf8 · 2025-06-02
> >
> > > ref: On Only Evaluating on a Programming Task
> >
> > I respect the opinions of other reviewers (including those from Reviewer p6Pc), but different viewpoints are natural in the review process, which is helpful for the ACs and PCs to make decisions. My role is to present my own evaluation based on my understanding:
> >
> > - As stated in my previous review, this paper claims that ideation and creative assistance are core targets of generation diversity (lines 23-25). Thus, rewriting the motivation would be a major revision of this paper and fundamentally alter the framework of this paper, which needs to be reviewed from scratch in another conference venue.
> >
> > - If the original motivation keeps unchanged, then, the ideation and creative assistance should be considered to be evaluated, as they are two of the most valuable applications of the generation diversity. But programming task is too weak to represent the ideation and creative assistance.
> >
> > Overall, I do not believe the authors are the ones who "start with a conclusion and retrofit the motivation", which is discouraged in the academic community. On the contrary, this paper may offer meaningful insights, but a broader evaluation across tasks should be added to validate the conclusions and insights in this paper.
> >
> > I still keep my mind open to adjust my assessment, if more experiment results well-address my concern above.
> >
> > > ref: On the potential overclaim on "eliminating the correlation with temperature"
> >
> > I have worried about this contribution is not actually exist, so I presented my concern about this point. But the authors' comments well-supported this contribution, and I have no more concern in this point.
> >
> > > ref: Re: What further insights can this metric bring to the design of reinforcement learning algorithms for LMs?
> >
> > Thanks for the explanation from the authors, and I suggest these points may be also valuable and could be added in a proper place in future revisions.

---

### Official Review · Reviewer_p6Pc · 2025-05-13

**Rating:** 8
**Confidence:** 4
**Ethics Flag:** 1

**Summary:**

The authors introduce Effective Semantic Diversity, a metric where an LLM generation counts toward diversity only if it (a) passes a simple functional-correctness check and (b) is semantically distinct from repeated samples (K).
They build a 108-prompt benchmark derived from CodeNet and AlphaCode, and sample several programs per prompt from dozens of Llama-family models  trained with SFT, DPO, PPO and GRPO, and test many temperature and nucleus-sampling variations. The authors promise to release a Docker runner, AST validator, and test-case generator, which is a nice touch but a bit future-looking.

I read this paper as an attempt to move the "diversity" discussion in LLM generation beyond the usual surface‐level token diversity measures. Some of the results are genuinely interesting, and the authors have put significant effort into the empirical work. A few citations for metrics/applications of diversity metrics in synthetic data are missing from the Introduction and Related work: [1-5]

[1] Yu et al. (NeurIPS 2023): "Large Language Model as Attributed Training Data Generator: A Tale of Diversity and Bias"

[2] Divekar et al. (EMNLP 2024): "SynthesizRR: Generating Diverse Datasets with Retrieval Augmentation"

[3] Chen et al. (http://arxiv.org/abs/2410.15226): "On the Diversity of Synthetic Data and its Impact on Training Large Language Models"

[4] Miranda et al. (https://arxiv.org/abs/2306.13840): "The diversity coefficient as a data quality metric for variability in natural language
data"

[5] Barnes et al. (ACL WASSA 2023): "ChatGPT is fun, but it is not funny! Humor is still challenging Large Language Models"

**Questions To Authors:**

1) Could you report even a tiny natural-language experiment? say, paraphrase generation with a grammar checker as the validity oracle to back the broader claims? This would give more confidence, and I am open to increasing my score during rebuttal if this was provided, as it directly counters my main issue with this work.

2) Have you tried stricter validity checks (e.g., unit tests with expected outputs) to see whether your diversity ranking is robust? A short experiment on a subset would suffice.

**Reasons To Accept:**

1) There's a strong clarity in the problem motivation, and ESD-like metrics are much-needed in long-form synthetic data. The authors nail why token-level diversity is unsatisfying, and offers a metric that rewards useful variety. I like that the "validity × diversity" framing because it is a simple tradeoff and easy to compute in domains where a runnable oracle exists. An additional metric like AUC would be useful for comparing models.

2) with 19 LLMs x four training regimes x three sampling schemes x non-parametric statistics, I think the authors have given the conclusions good grounding in experiments. The ablations identify where diversity gains origninate from.

3) While the core contribution is analysis-based, the appendix provides a stability property for the metric, which which is reassuring as it means rankings probably will not flip erratically with more samples. It's a good attempt to put the ESD metric on a theoretical footing.


4) The finding that small models are parameter-efficient at producing unique valid programs is a good fit for the "Technological Impact" dimension. It can be expanded a bit more.

**Reasons To Reject:**

1) My biggest issue with this paper is the framing of it. All evidence comes from code generation. It is too restricted in domain to make such general claims about the goodness of certain RL approaches. The paper repeatedly hints that the insights will transfer to other open-ended tasks, but there are no tests of these claim. Without at least a pilot natural-language study, it's hard to be convinced. One option is to reframe the paper to focus on code generation...this is a more limited but still interesting contribution and I think would stand on its own.

2) It would be good to see a small experiment on a different domain to back the broader claims. The authors could run a paraphrase generation task with a grammar checker as the validity oracle, for example.

3) Validity oracle is a bit weak for code. A program passes if it runs without error and prints anything... there's many situation where degenerate code can be "valid". Several prompts in the appendix look vulnerable to this loophole. This will artificially inflating ESD.

4) Needs more transparency on computing costs of method for reproducibility, as CoLM guidelines explicitly ask reviewers to weigh accessibility of compute (hardware specs, GPU-hours, etc).

5) Minor typos / wording issues:
L82: "undersirable" → undesirable
L160: Section heading "Semantic funtion" → function
L240–241: "the the additional results"
L137–138: "more / less diverse diverse than …"
Fig. 5 caption (L559) "semantic diveresity" → diversity
L559–563: Figure 5 caption begins with a lowercase clause and no period
L 71–73:   Sentence beginning "Additionally, in an attempt to understand [...]" ends abruptly.
L195–197: "1 N.B." inside main text interrupts flow, can be a footnote
(Multiple places): Phrases like "very clear trend" or "critical moment" feels super journalistic. More careful language and quantitative results (i.e. numbers from expts section) would help the results speak for themselves.

---

> ### Author Response · Authors · 2025-06-01
>
> Thank you for reading our paper and making the time to provide a thoughtful review! The related work you've provided is great and we look forward to reviewing them and including them!
>
> **Re: Validity oracle is a bit weak for code. A program passes if it runs without error and prints anything**
>
> Good catch! The correct description for how we do things is on lines 553-557: for all experiments *we evaluate only the returned object* and any spurious or intermediate print statements have no effect on the result. There's a discrepancy in the paper, because we did not properly update the prompts in the appendix. That's on us! We went through an iteration where we had this exact same concern as you: we were concerned about spurious print statements inflating ESD. We meant to update the prompts in the appendix to what we already have in the .zip supplementary material in the prompt_templates/open_ended*.txt files which read:
>
> ```
> Now please implement the function f. The function f will return the result of the operation. The function should return something in all cases.
> It should terminate within 30 seconds.
> ```
>
> additionally `utils/clustering/open_ended_wrapper.py` in the supplementary materials has a redacted version of the wrapper code used for executing and serializing the program result: it only considers the output of the completed function f and it serializes it to a file, thereby ignoring all intermediate print statements.
>
> We cannot update the PDF during rebuttal, but please know that we've already implemented this stricter method in all our experiments. That being said, we can do a tiny proof-of-concept where the tasks are the same, but we instruct and enforce a constraint to return a certain type: such a list of integers where the list must be shorter than 1000 elements.
>
> **Re: Issue with the framing of the paper**
>
> I can see how the framing of the paper leaves wanting for general results that go beyond what we have in code. I think we understand your perspective that either of two things can really happen: either we shift the language to qualify results for the domain of code ... or we try to provide a pilot study w.r.t. natural language. We think both may be appropriate. All three reviewers felt the focus on code was a significant point, and so we need to do a better job on our end either adding some supplementary experiments or adjusting the paper's language. And if we can provide some pilot study, we think that would be super cool.
>
> We cannot guarantee we'll be able to get the pilot study done, but what are your thoughts on using an LLM like GPT-4.1 as a judge for both diversity/similarity and either GPT-4.1 or a reward model for quality. Then for prompts, we could use either creative or argumentative essay prompts. I understand the suggestion of paraphrase generation, but we feel that domains like creative writing or essay-writing may be more open-ended and lend themselves better for semantic diversity measurement.
>
> On another note: COLM does not allow us to update the PDF, but in either case we would like to modify the language in our paper regarding your feedback. We'd like to provide some assurances on what we could do, but we're uncertain of what would be best.
>
> Just for some background context on why we emphasized code: in doing this work, we felt in a hard position, because we were uncertain about an automatic oracle for quality and diversity in natural language and we didn't want to open a paper up for criticism on that axis, hence we leaned into seeing how deep and far we could go with the code experiments. We anticipated future work to address this gap.
>
> **Re: Computing Cost Transparency and Language**
>
> Thank you for this feedback! After we looked back at the paper having had some time away from it, we can spot some typos and we totally agree there are some sentences / paragraphs where the writing can be more academic and less editorial and we'd like to address those. While we can't update the PDF please know we'll make those changes. Additionally, during the rebuttal window we'll provide the information for computing cost transparency.
>
> **Re specific questions:**
>
> 1. We're going to try our best to run some natural language experiments as discussed above.
> 2. As we mentioned, our implementation already uses a stricter validity check, AND we will also try to provide a tiny-experiment with even more constrained outputs as a proof of concept.

---

> > ### Author Response · Authors · 2025-06-10
> >
> > We wanted to follow up on our initial reply to you.
> >
> > ### On the Pilot Natural Language Study / Alternative Domain
> >
> > As we posted in the response to all reviewers at the top, we were able to run 3 different natural language tasks following the approach we took in our paper. Thank you for the suggestion to attempt some natural language experiments, and we hope you find our approach appropriate.
> >
> > ### On the Validity oracle
> >
> > As we mentioned in our original reply, the actual validity oracle we used in the paper did not allow spurious print statements to contribute to diversity: rather, we only considered the result returned by the function. Nevertheless, we also promised that we would run a proof of concept to demonstrate how we could further constrain generation with a more strict validity oracle
> >
> > As we originally suggested, we implemented a validity oracle that only accepts lists of integers of length 1000 or less. These are the exact lines of code we’ve added to enforce these constraints in our code execution driver; here, output is the result of running the LLM-generated function on the function’s inputs.
> >
> > ```
> > assert isinstance(output, list)
> > assert all(isinstance(x, int) for x in output)
> > assert len(output) < 1000
> > ```
> >
> > We also modified all our prompts to appropriately instruct program generations to conform with this change. Below is the table from running this constrained validity-oracle experiment. Generally we find highly similar results to those in our paper. Perhaps the most salient difference to us is the increase in significance and effect size of semantic diversity for DPO-tuned models relative to RL-tuned models for code. We believe this is all interesting, yet also does not impact major themes in the paper.
> >
> > ### Constrained Generation Results
> >
> > | Comparison | Validity (Quality) ||| Semantic Diversity ||| Syntactic Diversity ||| Lexical Diversity ||| Raw Neural Diversity |||
> > |---|---|---|---|---|---|---|---|---|---|---|---|---|---|---|---|
> > || **W (p)** | **ES (d)** | **Direction** | **W (p)** | **ES (d)** | **Direction** | **W (p)** | **ES (d)** | **Direction** | **W (p)** | **ES (d)** | **Direction** | **W (p)** | **ES (d)** | **Direction** |
> > | Base vs. Inst. | <0.001 | 1.23 | Inst. higher | <0.001 | 1.27 | Inst. higher | <0.001 | -0.88 | Base higher | 0.701 | -0.16 | Base higher | 0.010 | -1.61 | Base higher |
> > | Small vs. Lg. | 0.001 | 0.35 | Lg. higher | 0.001 | 0.35 | Lg. higher | 0.248 | 0.14 | Lg. higher | 0.645 | 0.02 | Lg. higher | 0.036 | 0.16 | Lg. higher |
> > | SFT vs. Pref. | <0.001 | 1.12 | Pref. higher | <0.001 | 1.04 | Pref. higher | <0.001 | -0.95 | SFT higher | <0.001 | -0.60 | SFT higher | <0.001 | -2.24 | SFT higher |
> > | SFT vs. DPO | 0.064 | 0.69 | DPO higher | 0.064 | 0.64 | DPO higher | <0.001 | -0.64 | SFT higher | 0.007 | -0.46 | SFT higher | <0.001 | -1.77 | SFT higher |
> > | SFT vs. RL | 0.004 | 3.28 | RL higher | 0.004 | 3.14 | RL higher | 0.004 | -2.18 | SFT higher | 0.008 | -1.04 | SFT higher | 0.004 | -2.98 | SFT higher |
> > | DPO vs. RL | 0.237 | -0.19 | DPO higher | 0.039 | -0.44 | DPO higher | 0.426 | -0.15 | DPO higher | 0.301 | -0.37 | DPO higher | 1.000 | 0.05 | RL higher |
> >
> >
> >
> > ### Transparency on computing costs of method for reproducibility
> >
> > For our work, we did our experiments on a server with 8 X 48GB Nvidia RTX A6000 and 96 Intel(R) Xeon(R) Gold 6342 (2.80GHz) CPUs where GPUs were our primary bottleneck. The experiments in the main manuscript take around 4.2 days of GPU hours on all 8 GPUs. Additionally, the natural language experiments took around 1.2 days to run. Lastly, because we repeated all the paper experiments for the constrained code generation experiment above, the total number of server days is around 9.5 days. We will include this information in a final paper draft. Additionally, the OpenAI API costs in order to do LLM-as-a-Judge evaluation for natural language were under $150 total. As we also explain in our natural language experiment writeup, a benefit of code execution as a quality and diversity oracle is that it avoids the additional costs of using potentially expensive LLMs for evaluation.
> >
> > ### Additional Remarks
> >
> > We’d like to thank you again for the time you’ve given in reviewing. As we mention above, we anticipate re-running at least one of the natural language experiments with an order-of-magnitude more prompts and adding the natural language experiments to a final draft. We appreciate the suggestion about AUC metrics; for a domain like natural language where we have soft diversity and quality scores, we will think about this further. We also look forward to improving our draft with the related papers you’ve shared; in fact we already do cite a version of the “ChatGPT is fun, but it is not funny!” paper on line 131, but we will update it to the ACL version you shared. We also will make the stylistic changes you’ve suggested and add this constrained-generation experiment to the paper’s appendix.

---

> > > ### Author Response · Authors · 2025-06-11
> > >
> > > Before the window closes, we'd just like to thank you for the time you have spent engaging with our paper. We truly appreciate it! It's clear you spent considerable time thinking about the paper and writing up your feedback. Your feedback really resonated with us, and the work will be much better for it. Reviewing takes time, and we're really grateful for the effort you gave.

---

### Author Response · Authors · 2025-06-10

Hi everyone, thank you for your patience while we tried to run experiments for natural language inspired by reviewer p6Pc’s suggestion. We believe these help address concerns about the generalizability of our findings. Tasks considered included creative writing, argumentative writing, and brainstorming. We believe our natural language results reflect many of our findings from code.

We obtained 10 prompts each for argumentative writing and creative writing using the CoAuthor dataset [1] which has been previously used in the literature [2]. Following reviewer 7jf8’s suggestion that “ideation and creative assistance” is important we also manually created a dataset of 10 brainstorming prompts that we think may reasonably reflect creative assistance / ideation tasks users may request from an LLM assistant. To evaluate diversity and quality we used GPT-4.1-mini as a judge to evaluate generations.

Overall we find that instruction-tuned models are associated with significantly higher Effective Semantic Diversity when compared with base models with effect sizes often ranging from large to very large. We also find in the creative writing and argumentative writing experiments that RL-tuned models are associated with significantly higher Effective Semantic Diversity relative to SFT-tuned models with their effect sizes ranging from large to very-large. We find weaker patterns in our brainstorming task; nevertheless, we believe all our results pose strong challenges to the dominant narrative that RLHF and post-training strategies in general reduce diversity.

While these natural language experiments provide valuable validation, we’d also like to emphasize that code generation offers unique advantages as an evaluation domain: execution-based validity checks and diversity comparisons are more objective than LLM-judge scores, computational costs are lower, and we avoid potential circularity of using LLMs to evaluate LLMs. For a final draft of this paper we plan to add the natural language experiments into the draft and increase the number of prompts for at least one of the sub-tasks by an order of magnitude.

# Results

Below we provide the tables modeled after Table 2 in our paper. As a reminder W(p) is the p-value of the Wilcoxon Signed-Rank Test and ES (d) is the Effect Size measured by Cohen’s D. Positive values indicate that the “right side” of the comparison is higher, but we included an extra column “Direction” to help clarify this further.

## Creative Writing Results

| Comparison | Validity (Quality) ||| Soft Effective Semantic Diversity ||| Hard Effective Semantic Diversity ||| Lexical Diversity ||| Raw Neural Diversity |||
|---|---|---|---|---|---|---|---|---|---|---|---|---|---|---|---|
|| **W (p)** | **ES (d)** | **Direction** | **W (p)** | **ES (d)** | **Direction** | **W (p)** | **ES (d)** | **Direction** | **W (p)** | **ES (d)** | **Direction** | **W (p)** | **ES (d)** | **Direction** |
| Base vs. Inst. | <0.001 | 1.69 | Inst. higher | <0.001 | 0.80 | Inst. higher | <0.001 | 1.67 | Inst. higher | <0.001 | 1.30 | Inst. higher | <0.001 | -3.21 | Base higher |
| Small vs. Lg. | <0.001 | 0.37 | Lg. higher | 0.004 | 0.47 | Lg. higher | 0.020 | 0.41 | Lg. higher | 0.202 | -0.13 | Small higher | 0.010 | -0.20 | Small higher |
| SFT vs. Pref. | <0.001 | 2.13 | Pref. higher | 0.010 | 0.62 | Pref. higher | 0.050 | 0.58 | Pref. higher | <0.001 | 0.95 | Pref. higher | <0.001 | -2.38 | SFT higher |
| SFT vs. DPO | <0.001 | 1.33 | DPO higher | 0.092 | 0.42 | DPO higher | 0.176 | 0.39 | DPO higher | <0.001 | 0.90 | DPO higher | <0.001 | -1.51 | SFT higher |
| SFT vs. RL | 0.004 | 5.80 | RL higher | 0.074 | 0.99 | RL higher | 0.203 | 0.83 | RL higher | 0.004 | 1.70 | RL higher | 0.004 | -5.57 | SFT higher |
| DPO vs. RL | 0.008 | 0.40 | RL higher | 0.129 | -0.45 | DPO higher | 0.203 | -0.29 | DPO higher | 0.570 | -0.15 | DPO higher | 0.098 | -0.37 | DPO higher |

---

> ### Author Response · Authors · 2025-06-10
>
> # Results (Continued)
>
> ## Argumentative Writing Results
>
> | Comparison | Validity (Quality) ||| Soft Effective Semantic Diversity ||| Hard Effective Semantic Diversity ||| Lexical Diversity ||| Raw Neural Diversity |||
> |---|---|---|---|---|---|---|---|---|---|---|---|---|---|---|---|
> || **W (p)** | **ES (d)** | **Direction** | **W (p)** | **ES (d)** | **Direction** | **W (p)** | **ES (d)** | **Direction** | **W (p)** | **ES (d)** | **Direction** | **W (p)** | **ES (d)** | **Direction** |
> | Base vs. Inst. | <0.001 | 1.93 | Inst. higher | <0.001 | 0.86 | Inst. higher | <0.001 | 1.69 | Inst. higher | 0.033 | 0.16 | Inst. higher | <0.001 | -3.07 | Base higher |
> | Small vs. Lg. | 0.248 | 0.12 | Lg. higher | 0.026 | 0.54 | Lg. higher | 0.044 | 0.37 | Lg. higher | 0.010 | -0.33 | Small higher | 0.594 | -0.00 | Small higher |
> | SFT vs. Pref. | <0.001 | 1.55 | Pref. higher | 0.033 | 0.49 | Pref. higher | 0.032 | 0.41 | Pref. higher | 0.919 | -0.21 | SFT higher | <0.001 | -1.90 | SFT higher |
> | SFT vs. DPO | <0.001 | 1.07 | DPO higher | 0.213 | 0.26 | DPO higher | 0.266 | 0.22 | DPO higher | 0.970 | -0.25 | SFT higher | <0.001 | -1.43 | SFT higher |
> | SFT vs. RL | 0.008 | 2.40 | RL higher | 0.129 | 1.07 | RL higher | 0.129 | 1.14 | RL higher | 0.910 | -0.14 | SFT higher | 0.004 | -3.11 | SFT higher |
> | DPO vs. RL | 0.020 | 0.22 | RL higher | 0.263 | 0.10 | RL higher | 0.301 | 0.12 | RL higher | 0.426 | 0.21 | RL higher | 0.359 | -0.20 | DPO higher |
>
> ## Brainstorming / Creative Assistance Results
>
> | Comparison | Validity (Quality) ||| Soft Effective Semantic Diversity ||| Hard Effective Semantic Diversity ||| Lexical Diversity ||| Raw Neural Diversity |||
> |---|---|---|---|---|---|---|---|---|---|---|---|---|---|---|---|
> || **W (p)** | **ES (d)** | **Direction** | **W (p)** | **ES (d)** | **Direction** | **W (p)** | **ES (d)** | **Direction** | **W (p)** | **ES (d)** | **Direction** | **W (p)** | **ES (d)** | **Direction** |
> | Base vs. Inst. | <0.001 | 1.27 | Inst. higher | <0.001 | 1.05 | Inst. higher | 0.002 | 0.86 | Inst. higher | <0.001 | 1.20 | Inst. higher | <0.001 | -3.16 | Base higher |
> | Small vs. Lg. | 0.859 | -0.32 | Small higher | 0.645 | -0.34 | Small higher | 0.546 | -0.23 | Small higher | 0.036 | -0.58 | Small higher | <0.001 | -0.48 | Small higher |
> | SFT vs. Pref. | 0.013 | 0.37 | Pref. higher | 0.137 | 0.15 | Pref. higher | 0.919 | -0.02 | SFT higher | 0.759 | -0.13 | SFT higher | <0.001 | -1.64 | SFT higher |
> | SFT vs. DPO | 0.339 | 0.14 | DPO higher | 0.970 | 0.01 | DPO higher | 0.151 | -0.10 | SFT higher | 0.176 | -0.31 | SFT higher | <0.001 | -1.32 | SFT higher |
> | SFT vs. RL | 0.020 | 0.72 | RL higher | 0.039 | 0.34 | RL higher | 0.426 | 0.09 | RL higher | 0.301 | 0.16 | RL higher | 0.008 | -2.19 | SFT higher |
> | DPO vs. RL | 0.055 | 0.24 | RL higher | 0.164 | 0.13 | RL higher | 0.203 | 0.12 | RL higher | 0.027 | 0.45 | RL higher | 0.734 | 0.36 | RL higher |
>
> ## Combined Results
>
> | Comparison | Validity (Quality) ||| Soft Effective Semantic Diversity ||| Hard Effective Semantic Diversity ||| Lexical Diversity ||| Raw Neural Diversity |||
> |---|---|---|---|---|---|---|---|---|---|---|---|---|---|---|---|
> || **W (p)** | **ES (d)** | **Direction** | **W (p)** | **ES (d)** | **Direction** | **W (p)** | **ES (d)** | **Direction** | **W (p)** | **ES (d)** | **Direction** | **W (p)** | **ES (d)** | **Direction** |
> | Base vs. Inst. | <0.001 | 1.49 | Inst. higher | <0.001 | 0.53 | Inst. higher | <0.001 | 0.78 | Inst. higher | <0.001 | 0.64 | Inst. higher | <0.001 | -1.97 | Base higher |
> | Small vs. Lg. | 0.010 | 0.07 | Lg. higher | 0.009 | 0.00 | Small higher | 0.054 | 0.05 | Lg. higher | <0.001 | -0.33 | Small higher | <0.001 | -0.15 | Small higher |
> | SFT vs. Pref. | <0.001 | 1.09 | Pref. higher | <0.001 | 0.15 | Pref. higher | 0.017 | 0.12 | Pref. higher | 0.084 | -0.05 | SFT higher | <0.001 | -1.20 | SFT higher |
> | SFT vs. DPO | <0.001 | 0.72 | DPO higher | 0.110 | 0.06 | DPO higher | 0.490 | 0.04 | DPO higher | 0.480 | -0.13 | SFT higher | <0.001 | -0.92 | SFT higher |
> | SFT vs. RL | <0.001 | 1.69 | RL higher | 0.003 | 0.27 | RL higher | 0.014 | 0.24 | RL higher | 0.055 | 0.09 | RL higher | <0.001 | -1.65 | SFT higher |
> | DPO vs. RL | <0.001 | 0.18 | RL higher | 0.590 | 0.04 | RL higher | 0.390 | 0.05 | RL higher | 0.069 | 0.24 | RL higher | 0.200 | -0.05 | DPO higher |

---

> > ### Author Response · Authors · 2025-06-10
> >
> > # Methodology and Datasets
> >
> > We evaluated the same exact cross section of models as those used in the paper and used zero-shot, two-shot, and two-shot with chain-of-thought directly reflecting how we did experiments for code. For each prompt in our dataset, we also took 32 samples as we did in our paper.
> >
> > ## Datasets
> >
> > As mentioned above, we use the Co-Author dataset for both creative writing and argumentative tasks. In the Co-Author dataset, the argumentative essays expected a human to respond to the prompt: because LLMs may be tuned to insist they are not human, we conservatively de-anthropomorphized the prompt (e.g. changing “In your opinion, what are the most important things students should learn in school” into “What are the most important things students should learn in school.”). In order to construct two-shot prompts, we used examples drawn from the NYT Editorial dataset [3] for the argumentative examples, and examples drawn from the WritingPrompts dataset [4]. Within our time constraints we could not find any “brainstorming” or “creative assistance” datasets, so we manually created 2 prompts for few-shot prompting and 10 for evaluation. Below are two examples for the brainstorming task:
> >
> > “Please provide one suggestion for a young professional living in a major global city on how they can get involved in community service or in giving back to the community in a meaningful way”
> >
> > “The manager of a remote workforce group of 20 employees is looking to improve team communication and cohesiveness. Please suggest one exercise or activity to take place between 3-6 hours (during overlapping work hours) on a Friday before the weekend in the Summer with the intention of helping improve team communication and cohesiveness.”
> >
> > ## Diversity and Quality Evaluation
> >
> > We used GPT-4.1-mini as a judge for both quality and diversity of generations. We chose GPT-4.1-mini to balance robustness with our API budget. We constructed separate prompts for each of the three tasks to outline the criteria for quality and diversity evaluation. For example here were the criteria for creative writing tasks:
> >
> > **Creative Quality Criteria:**
> >
> > > Consider the following rubric criteria while evaluating:
> > > 1. Overall/holistic/cohesive readability of the story (not just a compilation of elements).
> > > 2. That the story is relevant to the prompt provided.
> > > 3. Use of key narrative elements - vocabulary choice, imagery, setting, themes, dialogue, characterisation, point of view.
> > > 4. Structural elements and presentation which reflect control of structural elements such as spelling, grammar, punctuation, paragraphing, and formatting.
> > > 5. Overall plot logic: hook, conflict, initial crisis, rising and falling action, denouement/resolution.
> > > 6. Creativity/innovation/originality/research—credibility, new knowledge, avoidance of cliché and derivative tropes.
> >
> > **Creative Diversity Criteria:**
> >
> > > Consider the following criteria while evaluating similarity:
> > > 1. Semantic Overlap: Do the responses share similar underlying themes, ideas, narrative elements, or emotional content?
> > > 2. Thematic Consistency: Do both responses explore similar themes or motifs?
> >
> > Our prompts instructed the model to give a score 1-10 for each element; i.e. the first prompt would have a max score of 60, and the second would have a max score of 20. We would then use OpenAI’s grammar-constrained decoding to enforce an integer result (as well as a chain-of-thought like reasoning element with the hope of improving robustness). We then would normalize our scores by the max-score available.
> >
> > For diversity tasks we would sub-sample 32 total pairs from all possible pairs with replacement and ask the LLM-judge to score the similarity of the two generations. Then our diversity score would be $1 - \\text{Sim}(g_i^j, g_i^k)$.
> >
> > ### Effective Semantic Diversity
> >
> > Given that we only could calculate a pairwise diversity metric for natural language, we used equation 3 in the paper to calculate effective semantic diversity across sub-sampled pairs. For thoroughness we used two methods to measure pairwise effective semantic diversity that we thought were reasonable that we thought would capture the same intuition.
> >
> >
> > Recall the pairwise diversity metric from Equation 3:
> > $$
> > ESD_{pair}(P_i) = \frac{1}{\binom{K}{2}} \sum_{j < k} d_{sem}(g_i^j, g_i^k)
> > $$
> > where $d_{\\text{sem}}: \\mathcal{G} \\times \\mathcal{G} \\to \\{0,1\\}$ is defined as:
> > $$
> > d_{\\text{sem}}(g_i^j, g_i^k) =
> > \\begin{cases}
> > 0 & \\text{if either generation is invalid} \\\\
> > 0 & \\text{if both valid and semantically identical} \\\\
> > 1 & \\text{if both valid and semantically distinct}
> > \\end{cases}
> > $$
> >
> > For natural language, we adapt this using:
> >
> > **Hard Thresholding:**
> > $$
> > d_{\\text{sem}}(g_i^j, g_i^k) = \\begin{cases}
> > LLM_{\\text{div}}(g_i^j, g_i^k) & \\text{if } LLM_{\\text{qual}}(g_i^j) > 0.5 \\text{ and } LLM_{\\text{qual}}(g_i^k) > 0.5 \\\\
> > 0 & \\text{otherwise}
> > \\end{cases}
> > $$

---

> > > ### Author Response · Authors · 2025-06-10
> > >
> > > ### Effective Semantic Diversity (Continued)
> > >
> > >
> > > **Soft Weighting:**
> > > $$d_{\\text{sem}}(g_i^j, g_i^k) = LLM_{\\text{div}}(g_i^j, g_i^k) \\times LLM_{\\text{qual}}(g_i^j) \\times LLM_{\\text{qual}}(g_i^k)$$
> > >
> > > We found the second equation more informative and less noisy. The first equation is more similar to what we used for code, as for code there was no soft criterion of validity. Using either of these did not generally affect the results, but it did have some effect on significance values likely due to the extra variance that the thresholding effect can incur. We include both in our tables.
> > >
> > > Unlike our code experiments where program execution provides clear validity thresholds, natural language evaluation requires LLM-judge scoring or other neural model use, which may introduce additional noise and potential biases.
> > >
> > >
> > > # Analysis and Discussion
> > >
> > > Our results for natural language generally reflect our findings for code. Across all experiments, we generally find that post-training is associated with higher effective semantic diversity than base models. And we also witness that RL-tuned models are generally associated with higher effective semantic diversity than SFT-tuned models, often in a more dramatic fashion than DPO-tuned models. We also find that in argumentative and creative writing, larger models generally score higher in effective semantic diversity than smaller models, but the pattern is more muddled in our creative brainstorming task. Lastly, in these natural language tasks we actually find little to no evidence of post-training strategies inducing some kind of lexical mode-collapse.
> > >
> > > We believe the intuition/takeaways from the paper also apply here. We find that without the important consideration of quality, Raw Neural Diversity (from the LLM-judge) is higher in less-aggressive post-training regimes (i.e. base models are higher than instruction-tuned models, SFT models are higher than RL-tuned models, and so on). Nevertheless, the consideration and weighting of quality can more than offset this effect, exactly as we saw in code. As a result, more aggressive post-training regimes like PPO are associated with higher effective semantic diversity.
> > >
> > > I would like to credit all reviewers for their feedback as we agree that natural language experiments help validate our approach. We would also like to caution that we should approach the use of LLM-as-a-judge for diversity carefully: while the results may strengthen some of the claims of this approach, we’d caution the community against blindly trusting LLMs to evaluate LLMs for diversity, as we have not validated their ability to reflect human judgment in open-ended tasks when it comes to both quality and diversity especially when generations may be off-distribution. While these LLM-judge results align with our code-based findings, we maintain the same methodological concerns about LLM-as-judge evaluation that we expressed in lines 111-121. Our personal view is that the use of code remains an interpretable and principled way to evaluate LLM generation diversity and we strongly urge the community to rigorously evaluate if LLM-judges are appropriate for measuring diversity.
> > >
> > > We also believe these experiments highlight some advantages to using code to evaluate effective semantic diversity. For code, our validity criteria (our threshold for quality) required syntactic correctness and an ability to run all test cases without any Runtime Errors; for natural language an LLM-judge score is less interpretable and may be prone to reward-hacking. Additionally, in our experience, code execution for diversity and quality terminated relatively quickly and thus was not costly; whereas, LLM-judge evaluation may be expensive for researchers to scale: we ourselves needed to be judicious about sample sizes to manage potentially high API costs. Lastly, in a regime where more powerful models are being produced, using code to evaluate the diversity/quality tradeoff may have advantages to using weaker models to evaluate potentially stronger models.
> > >
> > > Again, we thank all reviewers for your careful review, suggestions, and the time you’ve given: we’ve certainly benefited from your honest feedback. For the final draft of the paper we plan to increase the number of prompts for either the creative writing task or the argumentative writing task or both by an order of magnitude and add these new results into the updated draft. Additionally, we would update our draft to reflect the additional findings and conclusions obtained from the NLP tasks under consideration.
> > >
> > > [1] Lee et al. "Coauthor: Designing a human-ai collaborative writing dataset for exploring language model capabilities." CHI 2022.
> > >
> > > [2] Padmakumar and He. "Does Writing with Language Models Reduce Content Diversity?" ICLR 2024.
> > >
> > > [3] He et al. "Decomposing argumentative essay generation via dialectical planning of complex reasoning." ACL Findings 2024.
> > >
> > > [4] Fan et al. "Hierarchical Neural Story Generation." ACL 2018.

---

> > > > ### Comment · Reviewer_p6Pc · 2025-06-10
> > > >
> > > > Thanks for the detailed reply and additional experiments. I think this addresses my main concerns. I have increased my score by 2.

---

> ### Comment · Reviewer_7jf8 · 2025-06-10
>
> Thanks for the response. I think the supplementary experimental results have addressed most of my concerns. I will raise my score.

---

### Author Response · Authors · 2025-06-11

We thank all reviewers for their thoughtful engagement throughout this process. In response to the primary concern about generalizability beyond code generation, we conducted additional natural language experiments across creative writing, argumentative writing, and brainstorming tasks. These experiments validate our key findings.

As of writing this, two reviewers (p6Pc and 7jf8) have increased their scores after reviewing our additional experiments. We’ve also addressed technical concerns about validity oracles with constrained generation experiments and provided computing cost transparency as requested.

For the final draft, we plan to:

- Include the natural language experiments with expanded sample sizes
- Expand the experimental results section
- Attempt to add syntactic analysis for natural language experiments
- Incorporate suggested citations and address stylistic feedback
- Add the constrained generation experiments to the appendix

We believe our work makes a valuable contribution to understanding the diversity-quality tradeoff in LLM generation, with practical implications for synthetic data generation and LLM evaluation. The code-based evaluation framework provides objective and scalable assessment while the natural language experiments demonstrate broader applicability.

---

### Decision · Program_Chairs · 2025-07-08

**Decision:**

Accept

**Comment:**

This paper evaluates the diversity of generated code, conditioned on generation quality (functional correctness), as well as some baseline diversity measures. They present several comparisons between different post-training approaches based on their proposed measure, and show that preference tuning might actually increase the “diversity” of generated code.

*Strengths*

- Brings into focus a critical aspect of generated code: its validity, even though the paper uses a weaker version of validity as pointed out by some reviewers. Regardless this emphasis on validity / diversity is valuable.
- The paper displays very strong empiricism, also evidenced in the author response.


*Weaknesses:*

- Given the emphasis on code generation, this paper will benefit from scoping its title and abstract better, are more narrowly. This is regardless of the additional empirical results in creative writing, where by the authors’ own admission: “domains like creative writing or essay-writing may be more open-ended and lend themselves better for semantic diversity measurement.”
- Some of the language in the paper seems somewhat non-standard for a scientific paper, this is something the authors promise to fix.

It would have been an easy decision if the paper did not make broader claims about all language, but with a narrower scoping will really benefit the paper.